# CHOICENET: ROBUST LEARNING BY REVEALING OUTPUT CORRELATIONS

## ABSTRACT

In this paper, we focus on the supervised learning problem with corrupt training data. We assume that the training dataset is generated from a mixture of a target distribution and other unknown distributions. We estimate the quality of each data by revealing the correlation between the generated distribution and the target distribution. To this end, we present a novel framework referred to here as ChoiceNet that can robustly infer the target distribution in the presence of inconsistent data. We demonstrate that the proposed framework is applicable to both classification and regression tasks. Particularly, ChoiceNet is evaluated in comprehensive experiments, where we show that it constantly outperforms existing baseline methods in the handling of noisy data in synthetic regression tasks as well as behavior cloning problems. In the classification tasks, we apply the proposed method to the MNIST and CIFAR-10 datasets and it shows superior performances in terms of robustness to different types of noisy labels.

## 1 INTRODUCTION

Training a deep neural network requires immense amounts of training data which are often collected using crowdsourcing methods, such as Amazon's Mechanical Turk (AMT). However, in practice, the crowd-sourced labels are often noisy (Bi et al., 2014). Furthermore, deep neural networks are vulnerable to over-fitting given the noisy training data in that they are capable of memorizing the entire dataset even with inconsistent labels, leading to a poor generalization performance (Zhang et al., 2016).

Assuming that a training dataset is generated from a mixture of a target distribution and other distributions, we address this problem through the principled idea of revealing the correlation between the target distribution and the other distributions. We present a framework for robust learning which is applicable to arbitrary neural network architectures such as convolutional neural networks (He et al., 2016a) or recurrent neural networks (Chung et al., 2014). We call this framework ChoiceNet.

Throughout this paper, we aim to address the following questions:

1. How can we measure the quality of training data in a principled manner?
2. In the presence of inconsistent outputs, how can we infer the target distribution in a scalable manner?

Traditionally, noisy outputs are handled by modeling additive random distributions, often leading to robust loss functions (Hampel et al., 2011). However, we argue that these approaches are too restrictive when handling severe outliers or inconsistencies in the datasets. To address the first question, we leverage the concept of a correlation. Precisely, we measure the quality of training data using the correlation between the target distribution and the data generating distribution. However, estimating the correct correlation requires an access to a target distribution, whereas learning the correct target distribution requires knowing the correlation between the distributions to be known, making it a chicken-and-egg problem. To address the second question, we simultaneously estimate the target distribution as well as the correlation in an end-to-end-manner using stochastic gradient decent methods, in this case Adam (Kingma & Ba, 2014), to achieve scalability.

The cornerstone of the proposed method is a mixture of correlated density network (MCDN) block. First, we present a Cholesky transform method for sampling the weights of a neural network that enables us to model correlated outputs. We also present an effective regularizer to train ChoiceNet.

To the best of our knowledge, this represents the first approach simultaneously to infer the target distribution and the output correlations using a neural network in an end-to-end manner.

Revealing the output correlations was proposed in earlier work (Bonilla et al., 2008), in which a multi-task Gaussian process prediction (MTGPP) model is proposed. In particular, MTGPP used correlated Gaussian processes to model multiple tasks by learning a free-form cross-covariance matrix. However, due to the multi-task learning setting, it is not suitable for learning a single target function. In other work (Choi et al., 2016), a leverage optimization method which optimizes the leverage of each demonstrations is proposed. Unlike a former study (Bonilla et al., 2008), the latter (Choi et al., 2016) focused on inferring a single expert policy by incorporating a sparsity constraint by assuming that the most demonstrations are collected from a skillful consistent expert. However, these methods suffer from the scalability issues when applying on large-scale tasks.

ChoiceNet is initially applied to a synthetic regression task, where we demonstrate its robustness to extreme outliers and ability to distinguish the target distribution and noise distributions. We then apply it to a behavior cloning scenario where the demonstrations are collected from both an expert and an adversarial policies. Subsequently, we move on to the classification tasks using the MNIST and CIFAR-10 datasets. We show that the proposed method outperforms existing baseline methods in terms of robustness with regard to the handling different types of noisy labels.

## 2 RELATED WORK

Recently, robustness in deep learning has been actively studied (Fawzi et al., 2017) as deep neural networks are being applied to diverse tasks involving real-world applications such as autonomous driving (Paden et al., 2016) or medical diagnosis (Gulshan et al., 2016) where a simple malfunction can have catastrophic results (AP & REUTERS, 2016).

Existing work for handing noisy training data can be categorized into four groups: small-loss tricks Jiang et al. (2017); Ren et al. (2018); Han et al. (2018); Malach & Shalev-Shwartz (2017), estimating label corruptions Patrini et al. (2017); Goldberger & Ben-Reuven (2017); Sukhbaatar et al. (2014); Bekker & Goldberger (2016); Hendrycks et al. (2018); Veit et al. (2017), using robust loss functions Natarajan et al. (2013); Belagiannis et al. (2015), and explicit and implicit regularization methods Reed et al. (2014); Lee (2013); Goodfellow et al. (2016); Xie et al. (2016); Tokozume et al. (2018); Zhang et al. (2017); Miyato et al. (2018); Tarvainen & Valpola (2017); Laine & Aila (2017). Our proposed method is mostly related to the robust loss function approach but cannot fully categorized into this group in that we present a novel architecture, a mixture of correlated density network block, for achieving robustness based on the correlation estimation.

First of all, the small-loss tricks selectively focus on training instances based on a certain criterion such as having small cost values Han et al. (2018). (Malach & Shalev-Shwartz, 2017) proposed a meta-algorithm for tackling the noisy label problem by training two networks only when the predictions of the two networks disagree, where selecting a proper network from among the two networks can be done using an additional clean dataset. (Ren et al., 2018) reweighs the weight of each training instance using a small amount of clean validation data. MentorNet Jiang et al. (2017) concentrated on the training of an additional neural network, which assigns a weight to each instance of training data to supervise the training of a base network, termed StudentNet, to overcome the over-fitting of corrupt training data. Recent work Han et al. (2018) presented Co-teaching by maintaining two separate networks where each network is trained with small-loss instances selected from its peer network.

The second group of estimating label corruption information is mainly presented for classification tasks where training labels are assumed to be corrupted with a possibly unknown corruption matrix. An earlier study in Bekker & Goldberger (2016) proposed an extra layer for the modeling of output noises. Jindal et al. (2016) extended the aforementioned approach by adding an additional noise adaptation layer with aggressive dropout regularization. A similar method was then proposed in Patrini et al. (2017) which initially estimated the label corruption matrix with a learned classifier and used the corruption matrix to fine-tune the classifier. Other researchers (Goldberger & Ben-Reuven, 2017) presented a robust training method that mimics the EM algorithm to train a neural network, with the label noise modeled as an additional softmax layer, similar to earlier work (Jindal et al., 2016). A self-error-correcting network was also presented (Liu et al., 2017). It switches the training labels based on the learned model at the beginning stages by assuming that the deep model is more accurate during the earlier stage of training.

Researchers have also focussed on using robust loss functions; (Natarajan et al., 2013) studied problem of binary classification in the presence of random labels and presented a robust surrogate loss function for handling noisy labels. Existing loss functions for classification were studied (Ghosh et al., 2017), with the results showing that the mean absolute value of error is inherently robust to label noise. In other work (Belagiannis et al., 2015), a robust loss function for deep regression tasks were proposed using Tukey's biweight function with median absolute deviation of the residuals.

The last group focusses on using implicit or explicit regularization methods while training. Adding small label noises while training is known to be beneficial to training, as it can be regarded as an effective regularization method (Lee, 2013; Goodfellow et al., 2016). Similar methods have been proposed to tackle noisy outputs. A bootstrapping method (Reed et al., 2014) which train a neural network with a convex combination of the output of the current network and the noisy target was proposed. Xie et al. (2016) proposed DisturbLabel, a simple method which randomly replaces a percentage of the labels with incorrect values for each iteration. Mixing both input and output data was also proposed (Tokozume et al., 2018; Zhang et al., 2017). One study (Zhang et al., 2017) considered the image recognition problem under label noise and the other (Tokozume et al., 2018) focused on a sound recognition problem. Temporal ensemble was proposed in Laine & Aila (2017) where an unsupervised loss term of fitting the output of of an augmented input to the augmented target updated with an exponential moving average. (Tarvainen & Valpola, 2017) extends the temporal ensemble in Laine & Aila (2017) by introducing a consistency cost function that minimizes the distance between the weights of the student model and the teacher model. (Miyato et al., 2018) presented new regularization method based on virtual adversarial loss which measures the smoothness of conditional label distribution given input. Minimizing the virtual adversarial loss has a regularizing effect in that it makes the model smooth at each data point.

The foundation of the proposed method is the mixture of correlated density network (MCDN) block where the output distribution is modeled using a mixture of correlated distributions. Modeling correlations of output training data has been actively studied in light of Gaussian processes (Rasmussen, 2006). MTGPP (Bonilla et al., 2008) that models the correlations of multiple tasks via Gaussian process regression was proposed in the context of multi-task setting. Choi et al. (2016) proposed a robust learning from demonstration method using a sparse constrained leverage optimization method which estimates the correlation between training outputs and showed its robustness compared to several baselines. On the other hand, Platanios et al. (2016) presented a graphical model that can model the structure of the error rates of learned classifiers using non-parametric Bayesian methods. We would like to note that while our problem setting is similar to the latter study (Choi et al., 2016), we propose end-to-end learning of both the target distribution and the correlation of each training data, thus offering, a clear advantage in terms of scalability.

## 3  CHOICENET

In this section, we present the methodology and the model architecture of ChoiceNet. A main ingredient of ChoiceNet is a mixture of correlated density network (MCDN) block upon the arbitrary base network. First, we illustrate the motivation of the MCDN block. Section 3.1 introduces Cholesky transform which enables correlated sampling and legitimates applying the reparametrization trick. Subsequently, we present the mechanism of ChoiceNet in Section 3.2 and loss functions for regression and classification tasks in Section 3.3.

**Modeling Corrupt Output**  As stated in Section 1, we focus on the problem of supervised learning on training data with corrupt outputs. Denote an unknown clean dataset by $\mathcal{D}_{\text{clean}}$ whose elements $(\mathbf{x}, y) \in \mathcal{D}_{\text{clean}}$ are determined by a relation $y = f(\mathbf{x})$. For a classification task, an accurate label $y \in \{0, 1\}$ exists for each $\mathbf{x}$. We assume a corrupt data $(\mathbf{x}, \hat{y}) \in \mathcal{D}_{\text{corrupt}}$ is given such that

**Regression**:           **Classification**:

$$\hat{y} = \begin{cases} f(\mathbf{x}) + \varepsilon & \text{with } 1 - p \\ g(\mathbf{x}) + \xi & \text{with } p \end{cases} \qquad \hat{y} = \begin{cases} y & \text{with } 1 - p \\ \{0, 1\} \setminus \{y\} & \text{with } p \end{cases}$$

where $g(\cdot)$ is an arbitrary function. Here $\varepsilon$ and $\xi$ are additive noise (usually heteroscedastic) and $p$ indicates the corruption (or mixture) probability. The above setting employs the random choice under Bernoulli distribution but one can consider a multinoulli distribution instead.

The corrupt data can be modeled by the conditional density estimation via a mixture of distributions:

$$\hat{y} \sim \pi_{\text{target}} P(\hat{y}|\mathbf{x}) + \pi_{\text{noise}} Q(\hat{y}|\mathbf{x})$$

where $\pi_{\text{target}}$ and $\pi_{\text{noise}}$ represent the ratio of target and noisy data, respectively. In this paper, we deal with the target conditional density $P(\cdot|\cdot)$ using a parametrized distribution with expected measurement variance $\hat{\sigma}^2$ i.e., $P(\cdot|\mathbf{x}) = \mathcal{N}(f_\theta(\mathbf{x}), \hat{\sigma}^2)$ where $f_\theta$ is a neural network and $\theta$ is a set of parameters. Analogous to the mixture density network (Bishop (1994)), we tackle the noise conditional distribution $Q(\cdot|\cdot)$ parametrized also by $\theta$. However, one major difference is that, we quantify its irrelevance (or independence) by utilizing the correlation $\rho$ between $P$ and $Q$. Intuitively speaking, irrelevant noisy data will be modeled to be collected from a class of $Q$ with relatively small or negative $\rho$. Since we assume that the correlation information is not explicitly given, we model the $\rho$ of each data to be a function of an input $\mathbf{x}$ i.e., $\rho_\phi(\mathbf{x})$, parametrized by $\phi$ and jointly optimize $\phi$ and $\theta$ using a mixture distribution. The MCDN block in Section 3.2 is proposed for this purpose.

## 3.1 CHOLESKY TRANSFORM AND CORRELATED SAMPLING

In this section, we present a novel method on how to model dependencies among output distributions. Given parameters $\Theta = (\rho, \mu, \sigma_1, \sigma_2)$, Cholesky transform is a mapping from $(w, z) \in \mathbb{R}^2$ to $\mathbb{R}$ which is defined by

$$\mathscr{T}_\Theta(w, z) := \rho\mu + \sqrt{1 - \rho^2} \left( \rho \frac{\sigma_2}{\sigma_1}(w - \mu) + z\sqrt{1 - \rho^2} \right)$$

Let $W$ and $Z$ be uncorrelated random variables such that $\mathbb{E}[W] = \mu_W$, $\mathbb{V}(W) = \sigma_W^2$, $\mathbb{E}[Z] = 0$, and $\mathbb{V}(Z) = \sigma_Z^2$. For $-1 \leq \rho \leq 1$, write $\tilde{\Theta} = (\rho, \mu_W, \sigma_W, \sigma_Z)$ and set a new random variable $\tilde{W}$ by plugging $\tilde{\Theta}$ and $(W, Z)$ in Cholesky transform i.e. $\tilde{W} := \mathscr{T}_{\tilde{\Theta}}(W, Z)$. This transform makes it possible to apply the reparametrization trick (Kingma (2017); Kingma & Welling (2013)) to jointly learn parameters not only $\mu_W, \sigma_W$ but also $\rho$. Since correlation is invariant to mean translation and variance dilatation (see appendix), it is easy to see that $\text{Corr}(W, \tilde{W}) = \rho$. The following theorem further states that a correlation between two random matrices is invariant to an affine transform.

**Theorem.** *Let $\boldsymbol{\rho} = (\rho_1, \ldots, \rho_K) \in \mathbb{R}^K$. For $p \in \{1, 2\}$, random matrices $\mathbf{W}^{(p)} \in \mathbb{R}^{K \times Q}$ are given such that for every $k \in \{1, \ldots, K\}$,*

$$\text{Cov}\left(W_{ki}^{(p)}, W_{kj}^{(p)}\right) = \sigma_p^2 \delta_{ij}, \quad \text{Cov}\left(W_{ki}^{(1)}, W_{kj}^{(2)}\right) = \rho_k \sigma_1 \sigma_2 \delta_{ij}$$

*Given $\mathbf{h} = (h_1, \ldots, h_Q) \in \mathbb{R}^Q$, set $\mathbf{y}^{(p)} = \mathbf{W}^{(p)}\mathbf{h}$ for each $p \in \{1, 2\}$. Then an elementwise correlation between $\mathbf{y}^{(1)}$ and $\mathbf{y}^{(2)}$ equals $\boldsymbol{\rho}$ i.e.*

$$\text{Corr}\left(y_k^{(1)}, y_k^{(2)}\right) = \rho_k, \quad \forall k \in \{1, \ldots, K\}$$

Hence $\boldsymbol{\rho}$ is a proper representation of dependencies among the distributions on the feature space. This theorem allows to use Cholesky transform to generate correlated weight matrices $\{\widetilde{\mathbf{W}}_k\}_{k=1}^K$ upon the feature layer of base network. In the following section, we demonstrate the details of the sampling process in the MCDN block.

## 3.2 MODEL ARCHITECTURE

In this section, we describe the model architecture and the overall mechanism of ChoiceNet. In the following, $\tau^{-1} > 0$ is a constant indicating expected measurement noise and $\eta(\cdot) \in (-1, 1)$ is a bounded function, e.g., a hyperbolic tangent. $\mathbf{W}_{\mathbf{h}\rightarrow\rho}, \mathbf{W}_{\mathbf{h}\rightarrow\boldsymbol{\pi}} \in \mathbb{R}^{K \times Q}$ and $\mathbf{W}_{\mathbf{h}\rightarrow\boldsymbol{\Sigma_o}} \in \mathbb{R}^{D \times Q}$ are weight matrices where $Q$ and $D$ denote the dimensions of a feature vector $\mathbf{h}$ and output $\mathbf{y}$, respectively, and $K$ is the number of mixtures. $\rho_{\max}$ is a fixed constant whose value is close to 1.

ChoiceNet is a twofold architecture: (a) a base network and (b) a MCDN block (see Figure 1). A base network extracts features for a given dataset. Then the MCDN block estimates the densities of the data generating distributions through $(\boldsymbol{\mu}_k, \boldsymbol{\Sigma}_k, \boldsymbol{\pi}_k)_{k=1}^K$. Contrary to the mixture density network (MDN), during the density estimation process, the MCDN block samples correlated weights $\{\widetilde{\mathbf{W}}_k\}_{k=1}^K$ using

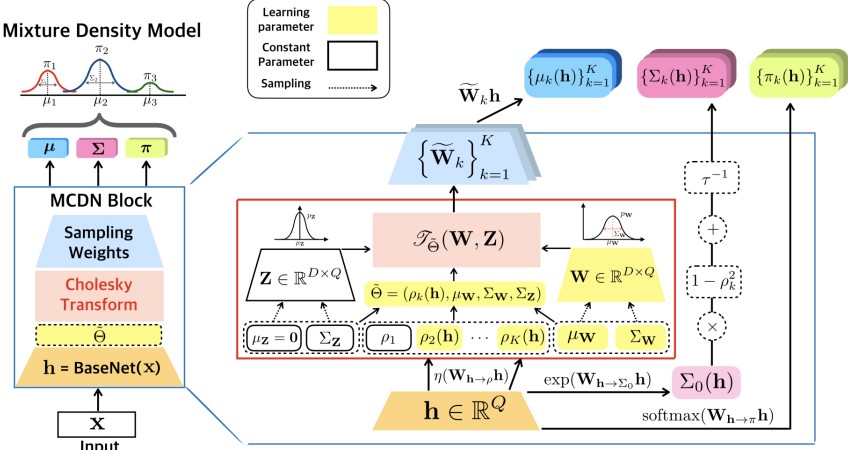

Figure 1: Model Architecture of ChoiceNet

Cholesky transform. Consequently, the MCDN block is able to model the correlated outputs i.e. correlated mean vectors $\boldsymbol{\mu}$. The overall mechanism of ChoiceNet can be elaborated as follows:

$$\text{Modules} = \begin{cases} \mathbf{h} = \text{BaseNet}(\mathbf{x}) \in \mathbb{R}^Q \\ \boldsymbol{\rho}(\mathbf{h}) = \eta(\mathbf{W}_{\mathbf{h}\to\rho}\mathbf{h}) \in \mathbb{R}^K = (\rho_1, \rho_2(\mathbf{h}), \dots, \rho_K(\mathbf{h})), \quad \rho_1 = \rho_{\max} \\ \Sigma_0(\mathbf{h}) = \exp(\mathbf{W}_{\mathbf{h}\to\Sigma_0}\mathbf{h}) \in \mathbb{R}^D \\ \Sigma_k = (1 - \rho_k^2)\Sigma_0(\mathbf{h}) + \tau^{-1} \in \mathbb{R}^D, \quad k \in \{1, \dots, K\} \end{cases}$$

$$\text{MCDN Block} = \begin{cases} \mathbf{W} \sim \mathcal{N}(\mu_{\mathbf{W}}, \Sigma_{\mathbf{W}}) \in \mathbb{R}^{D\times Q} \\ \mathbf{Z} \sim \mathcal{N}(\mathbf{0}, \Sigma_{\mathbf{Z}}) \in \mathbb{R}^{D\times Q} \\ \widetilde{\mathbf{W}}_k = \mathscr{T}_{(\rho_k(\mathbf{h}), \mu_{\mathbf{W}}, \Sigma_{\mathbf{W}}, \Sigma_{\mathbf{Z}})}(\mathbf{W}, \mathbf{Z}) \in \mathbb{R}^{D\times Q}, \quad k \in \{1, \dots, K\} \end{cases}$$

$$\text{Outputs} = \begin{cases} \boldsymbol{\mu} = (\mu_1, \dots, \mu_K) = (\widetilde{\mathbf{W}}_1\mathbf{h}, \dots, \widetilde{\mathbf{W}}_K\mathbf{h}) \in \mathbb{R}^{K\times D} \\ \boldsymbol{\pi} = (\pi_1, \dots, \pi_K) = \text{softmax}(\mathbf{W}_{\mathbf{h}\to\boldsymbol{\pi}}\mathbf{h}) \in \mathbb{R}^K \\ \boldsymbol{\Sigma} = (\Sigma_1, \dots, \Sigma_K) \in \mathbb{R}^{K\times D} \end{cases}$$

Thanks to the theorem in Section 3.1, for each $k \in \{1, \dots, K\}$

$$\text{Corr}(\mu_k, \mathbf{W}\mathbf{h}) = \text{Corr}(\widetilde{\mathbf{W}}_k\mathbf{h}, \mathbf{W}\mathbf{h}) = (\rho_k, \dots, \rho_k) \in \mathbb{R}^D$$

and the output density is modeled via mean vectors $\boldsymbol{\mu}$. Note that both $\mathbb{V}(\mu_k)$ and $\Sigma_k$ are minimized, when $\rho_k \to \pm 1$. Furthermore, as we employ Gaussian distributions in Cholesky transform, the influences of uninformative or independent data, whose correlations are close to 0, is attenuated as their variances increase (Kendall & Gal (2017)).

## 3.3 Training Objectives

Denote a training dataset by $\mathcal{D} = \{(\mathbf{x}_i, \mathbf{y}_i) : i = 1, \dots, N\}$. We consider both regression and classification tasks.

**Regression**    For the regression task, we employ both $L_2$-loss and the standard MDN loss (Bishop (1994); Choi et al. (2017); Christopher (2016))

$$\mathcal{L}(\mathcal{D}) = \frac{1}{N} \sum_{i=1}^{N} \left[ \lambda_1 \|\mathbf{y}_i - \mu_1(\mathbf{x}_i)\|_2^2 + \lambda_2 \log \left( \sum_{k=1}^{K} \pi_k(\mathbf{x}_i)\mathcal{N}(\mathbf{y}_i; \mu_k(\mathbf{x}_i), \text{diag}(\Sigma_k(\mathbf{x}_i))) \right) \right] \quad (1)$$

where $\lambda_1$ and $\lambda_2$ are hyper-parameters and $\mathcal{N}(\cdot|\mu, \Sigma)$ is the density of multivariate Gaussian:

$$\mathcal{N}(\mathbf{y}_i; \mu_k(\mathbf{x}_i), \text{diag}(\Sigma_k(\mathbf{x}_i))) = \prod_{d=1}^{D} \frac{1}{\sqrt{2\pi\Sigma_k^{(d)}}} \exp\left( -\frac{|y_i^{(d)} - \mu_k^{(d)}|^2}{2\Sigma_k^{(d)}} \right)$$

We also add weight decay and the following Kullback-Leibler regularizer to equation 1

$$\mathbb{KL}(\bar{\boldsymbol{\rho}}\|\boldsymbol{\pi}) = \sum_{k=1}^{K} \bar{\rho}_k \log \frac{\bar{\rho}_k}{\pi_k}, \quad \bar{\boldsymbol{\rho}} = \text{softmax}(\boldsymbol{\rho}) \tag{2}$$

The above KL regularizer encourages the mixture components with the strong correlations to have high mixture probabilities. This guidance is useful since ChoiceNet uses the mean vector $\mu_1(\mathbf{x}_i)$ of the first mixture component at the inference stage.

**Classification** In the classification task, we suppose each $\mathbf{y}_i$ is a $D$-dimensional one-hot vector. Unlike the regression task, equation 1 is not appropriate for the classification task. We employ the following loss function:

$$\mathcal{L}(\mathcal{D}) = -\frac{1}{N} \sum_{i=1}^{N} \sum_{k=1}^{K} \pi_k(\mathbf{x}_i) \left( \langle \text{softmax}(\hat{\mathbf{y}}_k(\mathbf{x}_i)), \mathbf{y}_i \rangle - \lambda_{\text{reg}} \log \left( \sum_{d=1}^{D} \exp(\hat{y}_k^{(d)}(\mathbf{x}_i)) \right) \right) \tag{3}$$

Here $\langle \cdot, \cdot \rangle$ denotes inner product and for $k \in \{1, \ldots, K\}, d \in \{1, \ldots, D\}$

$$\hat{\mathbf{y}}_k = (\hat{y}_k^{(1)}, \ldots, \hat{y}_k^{(D)}), \quad \hat{y}_k^{(d)}(\mathbf{x}_i) = \mu_k^{(d)} + \sqrt{\Sigma_k^{(d)}} \varepsilon, \quad \varepsilon \sim \mathcal{N}(0, 1)$$

Similar to the regression task, we use both equation 2 and weight decay.

# 4 EXPERIMENTS

## 4.1 REGRESSION TASKS

We conduct two regression experiments: 1) a synthetic scenario where the training dataset contains outliers sampled from other distributions and 2) a behavior cloning scenario where the demonstrations are collected from both expert and adversarial policies.

**Synthetic Regression Example** We first apply ChoiceNet to a simple one-dimensional regression problem of fitting $f(x) = \cos(\frac{\pi}{2}x) \exp(-(\frac{x}{2})^2)$ where $x \in [-3, +3]$ as shown in Figure 2. ChoiceNet is compared with a naive multilayer perceptron (MLP), a mixture density network (MDN) with five mixtures where all networks have two hidden layers with 32 nodes with a ReLU activation function. Gaussian process regression (GPR) (Rasmussen, 2006), leveraged Gaussian process regression (LGPR) with leverage optimization (Choi et al., 2016), and robust Gaussian process regression (RGPR) with an infinite Gaussian process mixture model (Rasmussen & Ghahramani, 2002) are also compared. For the GP based methods, we use a squared-exponential kernel function and the hyper-parameters are determined using a simple median trick (Dai et al., 2014)[1]. To evaluate its performance in corrupt datasets, we randomly replace the original target values with outliers whose output values are uniformly sampled from $-1$ to $+3$. We vary the outlier rates from $0\%$ (clean) to $80\%$ (extremely noisy).

Table 1 illustrates the RMSEs (root mean square errors) between the reference target function and the fitted results of ChoiceNet and other compared methods. Given an intact training dataset, all the methods show stable performances in that the RMSEs are all below $0.1$. Given training datasets whose outlier rates exceed $40\%$, however, only ChoiceNet successfully fits the target function whereas the other methods fail as shown in Figure 2.

To further inspect whether ChoiceNet can distinguish between the target distribution and noise distributions, we train ChoiceNet on two datasets. In particular, we use the same target function and replace $50\%$ of the output values whose input values are within 0 to 2 using two different corruptions: one uniformly sampled from $-1$ to 3 and the other from a flipped target function. For this experiment, we set $K = 2$ for better visualization. As shown in Figure 3(a) and 3(c), ChoiceNet successfully fits the target function. The correlations of the second component decrease as outliers are introduced as shown in Figure 3(b) and 3(d). Surprisingly, when the target and noise distribution are negatively correlated (the flipped function case), the correlations of the second component become $-1$ as depicted in Figure 3(b). Contrarily, for the uniform corruption case, the correlations of the second component are within 0 and 1. We would like to emphasize that this clearly shows the capability of ChoiceNet to distinguish the target distribution from noisy distributions.

---

[1] A median trick selects the length parameter of a kernel function to be the median of all pairwise distances between training data.

Table 1: RMSE of compared methods on synthetic toy examples

| Outliers | ChoiceNet | MDN | MLP | GPR | LGPR | RGPR |
|---|---|---|---|---|---|---|
| 0% | 0.034 | 0.028 | 0.039 | **0.008** | 0.022 | 0.017 |
| 20% | 0.022 | 0.087 | 0.413 | 0.280 | 0.206 | **0.013** |
| 40% | **0.018** | 0.565 | 0.452 | 0.447 | 0.439 | 1.322 |
| 60% | **0.023** | 0.645 | 0.636 | 0.602 | 0.579 | 0.738 |
| 80% | **0.084** | 0.778 | 0.829 | 0.779 | 0.777 | 1.523 |

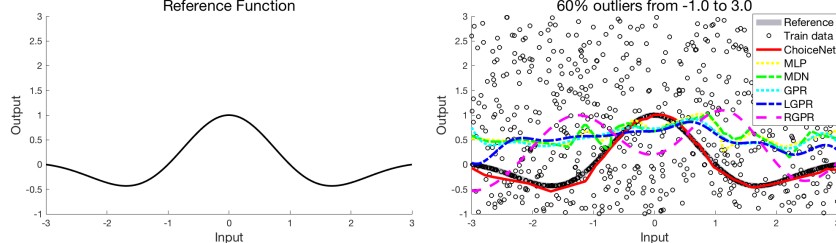

Figure 2: Reference function and fitting results of compared methods on different outlier rates.

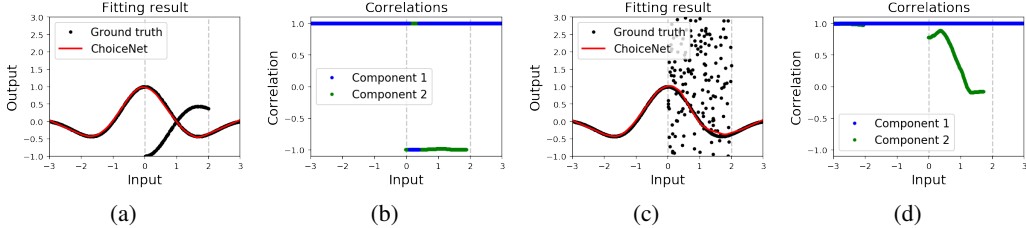

Figure 3: Fitting results on datasets with (a) flipped function and (c) uniform corruptions. Resulting correlations of two components with (b) flipped function and (d) uniform corruptions.

Table 2: Average returns of compared methods on MuJoCo problems

| Outliers | HalfCheetah | | | Walker2d | | |
|---|---|---|---|---|---|---|
| | ChoiceNet | MDN | MLP | ChoiceNet | MDN | MLP |
| 10% | **2068.14** | 192.53 | 852.91 | **2754.08** | 102.99 | 537.42 |
| 20% | **1498.72** | 675.94 | 372.90 | **1887.73** | 95.29 | 1155.80 |
| 30% | **2035.91** | 363.08 | 971.24 | -267.10 | **-260.80** | -728.39 |

**Behavior Cloning Example** In this experiment, we apply ChoiceNet to behavior cloning tasks when given demonstrations with mixed qualities. where the proposed method is compared with MLP and MDN in two locomotion tasks: *HalfCheetah* and *Walker2d*. The network architectures are identical to those in the synthetic regression example tasks. To evaluate the robustness of ChoiceNet, we collect demonstrations from both an expert policy and an adversarial policy where two policies are trained by solving the corresponding reinforcement learning problems using the state-of-the-art proximal policy optimization (PPO) (Schulman et al., 2017). For training adversarial policies for both tasks, we flip the signs of the directional rewards so that the agent gets incentivized by going backward. We evaluate the performances of the compared methods using 500 state-action pairs with different mixing ratio and measure the average return over 100 consecutive episodes. The results are shown in Table 2. In both cases, ChoiceNet outperforms compared methods by a significant margin.

## 4.2 CLASSIFICATION TASKS

We also conduct classification experiments on the MNIST and CIFAR-10 datasets to evaluate the performance of ChoiceNet on corrupt labels. To generate noisy datasets, we follow the setting in

(Zhang et al., 2017) which randomly shuffles a percentage of the labels in the dataset[2]. We vary the corruption probabilities from $50\%$ to $95\%$ for the MNIST dataset and from $20\%$ to $80\%$ for the CIFAR-10 dataset and compare median accuracies after five runs for each configuration. On both MNIST and CIFAR-10 experiments, we also compare ChoiceNet with Mixup (Zhang et al., 2017) which, to the best of our knowledge, shows the state-of-the-art performance on noisy labels. We set the parameter $\alpha$ of Mixup to be $32$ for the baseline network as suggested in the original paper. For ChoiceNet, we set $\alpha$ to be $1$.

For the MNIST experiments, we construct two networks: a network with two residual blocks (He et al., 2016b) with $3 \times 3 \times 64$ convolutional layers followed by a fully-connected layer with $256$ output neurons (ConvNet) and a network with the same two residual blocks followed by a MCDN block (ChoiceNet). We train each network for $50$ epochs with a fixed learning rate of $1e-5$. For the CIFAR experiments, we adopt WideResNet (WRN) (Zagoruyko & Komodakis, 2016) with $22$ layers and a widening factor of $4$. To construct ChoiceNet, we replace the last layer of WideResNet with a MCDN block. We set $K = 3$, $\rho_{\max} = 0.95$, $\lambda_{\mathrm{reg}} = 0.0001$, and $\rho_k, \pi_k, \Sigma_0$ modules consist of two fully connected layers with $64$ hidden units and a ReLU activation function. We train each network for $300$ epochs with a minibatch size of $256$. We begin with a learning rate of $0.1$, and it decays by $1/10$ after $150$ and $225$ epochs. We apply random horizontal flip and random crop with $4-$pixel-padding and use a weight decay of $0.0001$ for the baseline network as (He et al., 2016b). However, to train ChoiceNet, we reduce the weight decay rate to $1e-6$ and apply gradient clipping at $1.0$. We also lower the learning rate to $0.001$ for the first epoch to stabilize training.

Table 3: Test accuracies on the MNIST datasets with corrupt labels.

| Corruption $p$ | Configuration | Best | Last |
|---|---|---|---|
| 50% | ConvNet | 95.4 | 89.5 |
| | ConvNet+Mixup | 97.2 | 96.8 |
| | ConvNet+CN | **99.2** | **99.2** |
| 80% | ConvNet | 86.3 | 76.9 |
| | ConvNet+Mixup | 87.2 | 87.2 |
| | ConvNet+CN | **98.2** | **97.6** |
| 90% | ConvNet | 76.1 | 69.8 |
| | ConvNet+Mixup | 74.7 | 74.7 |
| | ConvNet+CN | **94.7** | **89.0** |
| 95% | ConvNet | 72.5 | 64.4 |
| | ConvNet+Mixup | 69.2 | 68.2 |
| | ConvNet+CN | **88.5** | **80.0** |

Table 4: Test accuracies on the CIFAR-10 datasets with corrupt labels

| Corruption $p$ | Configuration | Best | Last |
|---|---|---|---|
| 20% | ConvNet | 88.5 | 85.3 |
| | ConvNet+CN | 90.7 | 90.3 |
| | ConvNet+Mixup | **92.9** | **92.3** |
| | ConvNet+Mixup+CN | 92.5 | **92.3** |
| 50% | ConvNet | 79.7 | 59.3 |
| | ConvNet+CN | 85.9 | 84.6 |
| | ConvNet+Mixup | 87.3 | 83.1 |
| | ConvNet+Mixup+CN | **88.4** | **87.9** |
| 80% | ConvNet | 67.8 | 27.4 |
| | ConvNet+CN | 69.8 | 65.2 |
| | ConvNet+Mixup | 72.1 | 62.9 |
| | ConvNet+Mixup+CN | **76.1** | **75.4** |

The classification results of the MNIST dataset and the CIFAR dataset are shown in Table 3 and Table 4, respectively. In the MNIST experiments, ChoiceNet consistently outperforms ConvNet and ConvNet+Mixup by a significant margin, and the difference between the accuracies of ChoiceNet and the others becomes more clear as the corruption probability increases. Particularly, the best test accuracy of ChoiceNet reaches $94\%$ even when $90\%$ of the training labels are randomly shuffled.

In the CIFAR-10 experiments, ChoiceNet outperforms WideResNet and achieves its accuracy over $60\%$ even when $80\%$ of the labels are shuffled whereas the accuracy of WideResNet drops below $30\%$. When we inspect the training accuracies on the $80\%$-shuffled set, WideResNet tends to overfit (memorize) to noisy labels and shows $99.8\%$ train accuracy. On the contrary, ChoiceNet shows $37.6\%$.[3] When trained with Mixup, both networks become robust to noisy labels to some extent. However, the results of the two networks still show significant differences except for the $20\%$ corrupt experiments on which both of them show similar accuracies. Interestingly, when ChoiceNet and Mixup are combined, it achieves a high accuracy of $75\%$ even on the $80\%$ shuffled dataset. We also

---

[2]In the corrupt label setting, for a given corruption probability $p$, the expected ratio of correct labels is $(1 - p) + p \times 1/(\text{number of classes})$. Additional experiments of replacing the percentage of labels to a random labels and a fixed label can be found in the appendix.

[3] Detailed learning curves can be found in the appendix.

note that ChoiceNet (without Mixup) outperforms WideResNet+Mixup when the corruption ratio is over $50\%$ on the last accuracies.

## 5 CONCLUSION

In this paper, we have presented ChoiceNet that can robustly learn a target distribution given noisy training data. The keystone of ChoiceNet is the mixture of correlated density network block which can estimate the densities of data distributions using a set of correlated mean functions. We have demonstrated that ChoiceNet can robustly infer the target distribution on corrupt training data in the following tasks; regression with synthetic data, behavior cloning using demonstrations with mixed qualities, and MNIST and CIFAR-10 image classification tasks. Our experiments verify that ChoiceNet outperforms existing methods in the handling of noisy data.

Selecting proper hyper-parameters including the optimal number of mixture components is a compelling topic for the practical usage of ChoiceNet. Furthermore, one can use ChoiceNet for active learning by evaluating the quality of each training data using through the lens of correlations. We leave these as important questions for future work.

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

## A   PROOF OF THEOREM IN SECTION 3

In this appendix, we introduce fundamental theorems which lead to Cholesky transform for given random variables $(W, Z)$. We apply this transform to random matrices $\mathbf{W}$ and $\mathbf{Z}$ which carry out weight matrices for prediction and a supplementary role, respectively. We also elaborate the details of conducted experiments with additional illustrative figures and results. Particularly, we show additional classification experiments with the MNST dataset on different noise configurations.

*Lemma* 1. Let $W$ and $Z$ be uncorrelated random variables such that

$$\begin{cases} \mathbb{E}W = \mu_W, & \mathbb{V}(W) = \sigma_W^2 \\ \mathbb{E}Z = 0, & \mathbb{V}(Z) = \sigma_Z^2 \end{cases} \tag{4}$$

For a given $-1 \leq \rho \leq 1$, set

$$\tilde{Z} = \rho \frac{\sigma_Z}{\sigma_W}(W - \mu_W) + \sqrt{1 - \rho^2} Z \tag{5}$$

Then $\mathbb{E}\tilde{Z} = 0$, $\mathbb{V}(\tilde{Z}) = \sigma_Z^2$, and $\text{Corr}(W, \tilde{Z}) = \rho$.

*Proof.* Since $W$ and $Z$ are uncorrelated, we have

$$\mathbb{E}\left[(W - \mu_W)Z\right] = \mathbb{E}(W - \mu_W)\mathbb{E}Z = 0 \tag{6}$$

By equation 4, we directly obtain

$$\mathbb{E}\tilde{Z} = \rho \frac{\sigma_Z}{\sigma_W}(\mathbb{E}W - \mu_W) + \mathbb{E}Z = 0$$

Also, by equation 4 and equation 6,

$$\mathbb{V}\left(\tilde{Z}\right) = \mathbb{E}|\tilde{Z}|^2 = \rho^2 \left(\frac{\sigma_Z}{\sigma_W}\right)^2 \mathbb{V}(W) + \mathbb{V}(Z) + 2\rho \frac{\sigma_Z}{\sigma_W} \underbrace{\mathbb{E}\left[(W - \mu_W)Z\right]}_{=0}$$

$$= \rho^2 \frac{\sigma_Z^2}{\sigma_W^2}\sigma_W^2 + (1 - \rho^2)\sigma_Z^2 = \sigma_Z^2$$

Similarly,

$$\text{Cov}(W, \tilde{Z}) = \mathbb{E}\left[(W - \mu_W)\tilde{Z}\right]$$

$$= \mathbb{E}\left[(W - \mu_W)\rho\frac{\sigma_Z}{\sigma_W}(W - \mu_W)\right] + \underbrace{\mathbb{E}\left[(W - \mu_W)Z\right]}_{=0}$$

$$= \rho\frac{\sigma_Z}{\sigma_W}\mathbb{V}(W) = \rho\sigma_Z\sigma_W$$

Therefore

$$\text{Corr}(W, \tilde{Z}) = \frac{\text{Cov}(W, \tilde{Z})}{\sqrt{\mathbb{V}(W)}\sqrt{\mathbb{V}(\tilde{Z})}} = \frac{\rho\sigma_W\sigma_Z}{\sigma_W\sigma_Z} = \rho$$

The lemma is proved. $\qquad\qquad\square$

*Lemma* 2. Assume the same condition in Lemma 1 and define $\tilde{Z}$ as equation 5. For given functions $\varphi : \mathbb{R} \to \mathbb{R}$ and $\psi : \mathbb{R} \to (0, \infty)$, set $\tilde{W} := \varphi(\rho) + \psi(\rho)\tilde{Z}$. Then

$$\mathbb{E}\tilde{W} = \varphi(\rho), \quad \mathbb{V}(\tilde{W}) = |\psi(\rho)|^2\sigma_Z^2, \quad \text{Corr}(W, \tilde{W}) = \rho$$

*Proof.* Note that

$$\mu_{\tilde{W}} = \varphi(\rho) + \psi(\rho)\mu_{\tilde{Z}} = \varphi(\rho)$$

$$\sigma_{\tilde{W}}^2 = |\psi(\rho)|^2\, \mathbb{E}\left(\tilde{Z} - \mu_{\tilde{Z}}\right)^2 = \psi^2(\rho)\sigma_Z^2$$

Therefore, by Lemma 1

$$\mathbb{E}\left[(W - \mu_W)(\tilde{W} - \mu_{\tilde{W}})\right] = \psi(\rho)\mathbb{E}\left[(W - \mu_W)(\tilde{Z} - \mu_{\tilde{Z}})\right]$$
$$= \rho\psi(\rho)\sigma_W\sigma_Z$$

Hence

$$\mathrm{Corr}(W, \tilde{W}) = \frac{\mathbb{E}\left[(W - \mu_W)(\tilde{W} - \mu_{\tilde{W}})\right]}{\sigma_W\sigma_{\tilde{W}}} = \frac{\rho\psi(\rho)\sigma_W\sigma_Z}{\psi(\rho)\sigma_W\sigma_Z} = \rho$$

The lemma is proved. $\qquad\qquad\square$

Now we prove the aforementioned theorem in Section 3.

**Theorem.** *Let $\boldsymbol{\rho} = (\rho_1, \ldots, \rho_K) \in \mathbb{R}^K$. For $p \in \{1, 2\}$, random matrices $\mathbf{W}^{(p)} \in \mathbb{R}^{K \times Q}$ are given such that for every $k \in \{1, \ldots, K\}$,*

$$\mathrm{Cov}\left(W_{ki}^{(p)}, W_{kj}^{(p)}\right) = \sigma_p^2 \delta_{ij}, \quad \mathrm{Cov}\left(W_{ki}^{(1)}, W_{kj}^{(2)}\right) = \rho_k\sigma_1\sigma_2\delta_{ij} \tag{7}$$

*Given $\mathbf{h} = (h_1, \ldots, h_Q) \in \mathbb{R}^Q$, set $\mathbf{y}^{(p)} = \mathbf{W}^{(p)}\mathbf{h}$ for each $p \in \{1, 2\}$. Then an elementwise correlation between $\mathbf{y}^{(1)}$ and $\mathbf{y}^{(2)}$ equals $\boldsymbol{\rho}$ i.e.*

$$\mathrm{Corr}\left(y_k^{(1)}, y_k^{(2)}\right) = \rho_k, \quad \forall k \in \{1, \ldots, K\}$$

*Proof.* First we prove that for $p \in \{1, 2\}$ and $k \in \{1, \ldots, K\}$

$$\mathbb{V}\left(y_k^{(p)}\right) = \sigma_p^2 \left\|\mathbf{h}\right\|^2 \tag{8}$$

Note that

$$\mathbb{V}\left(y_k^{(p)}\right) = \mathbb{E}\left[\left(\sum_{i=1}^Q W_{ki}^{(p)}h_i - \mathbb{E}\left[\sum_{i=1}^Q W_{ki}^{(p)}h_i\right]\right)^2\right]$$
$$= \mathbb{E}\left[\left(\sum_{i=1}^Q \left(W_{ki}^{(p)} - \mathbb{E}W_{ki}^{(p)}\right)h_i\right)^2\right]$$
$$= \mathbb{E}\left[\sum_{i,j}^Q \left(W_{ki}^{(p)} - \mathbb{E}W_{ki}^{(p)}\right)\left(W_{kj}^{(p)} - \mathbb{E}W_{kj}^{(p)}\right)h_ih_j\right]$$
$$= \sum_{i,j}^Q \mathrm{Cov}(W_{ki}^{(p)}, W_{kj}^{(p)})h_ih_j$$

By equation 7,

$$\mathbb{V}\left(y_k^{(p)}\right) = \sum_{i,j}^Q \mathrm{Cov}(W_{ki}^{(p)}, W_{kj}^{(p)})h_ih_j = \sum_{i,j}^Q \sigma_p^2 h_ih_j\delta_{ij} = \sum_{i=1}^Q \sigma_p^2 h_i^2 = \sigma_p^2\|\mathbf{h}\|^2$$

so equation 8 is proved. Next we prove

$$\mathrm{Cov}(y_k^{(1)}, y_k^{(2)}) = \rho_k\sigma_1\sigma_2 \left\|\mathbf{h}\right\|^2 \tag{9}$$

Observe that

$$
\begin{aligned}
\mathrm{Cov}(y_k^{(1)}, y_k^{(2)}) &= \mathbb{E}\left[\left(y_k^{(1)} - \mathbb{E}y_k^{(1)}\right)\left(y_k^{(2)} - \mathbb{E}y_k^{(2)}\right)\right] \\
&= \mathbb{E}\left[\left(\sum_{i=1}^{Q} W_{ki}^{(1)} h_i - \mathbb{E}\left[\sum_{i=1}^{Q} W_{ki}^{(1)} h_i\right]\right)\left(\sum_{j=1}^{Q} W_{kj}^{(2)} h_j - \mathbb{E}\left[\sum_{j=1}^{Q} W_{kj}^{(2)} h_j\right]\right)\right] \\
&= \mathbb{E}\left[\sum_{i,j}^{Q}\left(W_{ki}^{(1)} - \mathbb{E}W_{ki}^{(1)}\right)\left(W_{kj}^{(2)} - \mathbb{E}W_{kj}^{(2)}\right) h_i h_j\right] \\
&= \sum_{i,j}^{Q} \mathrm{Cov}(W_{ki}^{(1)}, W_{kj}^{(2)}) h_i h_j
\end{aligned}
$$

Similarly,

$$
\mathrm{Cov}(y_k^{(1)}, y_k^{(2)}) = \sum_{i,j}^{Q} \mathrm{Cov}(W_{ki}^{(1)}, W_{kj}^{(2)}) h_i h_j = \sum_{i,j}^{Q} \rho_k \sigma_1 \sigma_2 h_i h_j \delta_{ij} = \rho_k \sigma_1 \sigma_2 \|\mathbf{h}\|^2
$$

Hence equation 9 is proved. Therefore by equation 8 and equation 9

$$
\mathrm{Corr}(y_k^{(1)}, y_k^{(2)}) = \frac{\mathrm{Cov}(y_k^{(1)}, y_k^{(2)})}{\sqrt{\mathbb{V}(y_k^{(1)})}\sqrt{\mathbb{V}(y_k^{(2)})}} = \frac{\rho_k \sigma_1 \sigma_2 \|\mathbf{h}\|^2}{\sqrt{\sigma_1^2 \|\mathbf{h}\|^2}\sqrt{\sigma_2^2 \|\mathbf{h}\|^2}} = \rho_k
$$

The theorem is proved. $\qquad \square$

*Remark.* Recall the definition of Cholesky transform: for $-1 < \rho < 1$

$$
\mathscr{T}_{(\rho, \mu_W, \sigma_W, \sigma_Z)}(w, z) := \rho \mu_W + \sqrt{1 - \rho^2}\left(\rho \frac{\sigma_Z}{\sigma_W}(w - \mu_W) + \sqrt{1 - \rho^2} z\right) \tag{10}
$$

Note that we do not assume $W$ and $Z$ should follow typical distributions. Hence every above theorems hold for general class of random variables. Additionally, by Theorem 2 and equation 10, $\tilde{W}$ has the following $\rho$-dependent behaviors;

$$
\mathbb{E}\tilde{W} \to \begin{cases} \mu_W & : \rho \to 1 \\ 0 & : \rho \to 0 \\ -\mu_W & : \rho \to -1 \end{cases}, \quad \mathbb{V}(\tilde{W}) \to \begin{cases} 0 & : \rho \to \pm 1 \\ \sigma_Z^2 & : \rho \to 0 \end{cases}
$$

Thus strongly correlated weights $\tilde{W}$ i.e. $\rho \approx 1$, provide prediction with confidence while uncorrelated weights encompass uncertainty. These different behaviors of weights perform regularization and preclude over-fitting caused by bad data since uncorrelated and negative correlated weights absorb vague and outlier pattern, respectively.

## B  ADDITIONAL EXPERIMENTS

### B.1  REGRESSION TASKS

#### B.1.1  SYNTHETIC EXAMPLE

We provide more fitting results for the synthetic example in Figure 4. Given an intact dataset, all compared methods robustly fit the given training data. However, other methods fail to correctly fit the underlying target function given corrupt data. When the outlier rate exceeds $90\%$ all tested methods fail to fit.

#### B.1.2  AUTONOMOUS DRIVING EXPERIMENT

**Autonomous Driving Experiment**  In this experiment, we apply ChoiceNet to a autonomous driving scenario in a simulated environment. In particular, the tested methods are asked to learn the

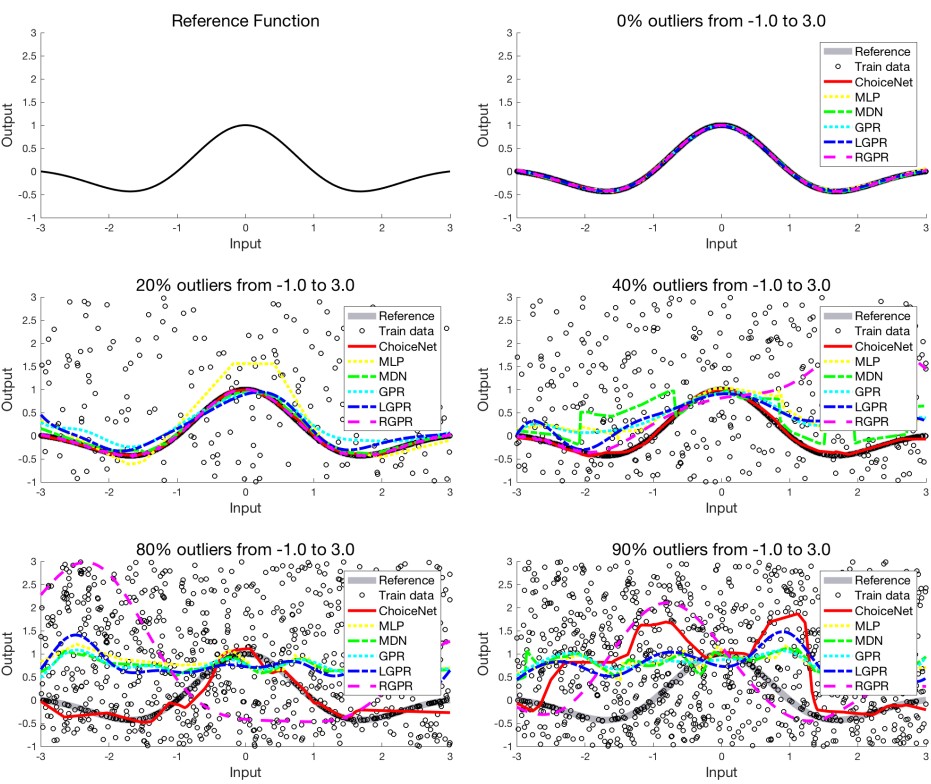

Figure 4: Reference function and fitting results of compared methods on different outlier rates, 0%,20% 40%, 80%, and 90%).

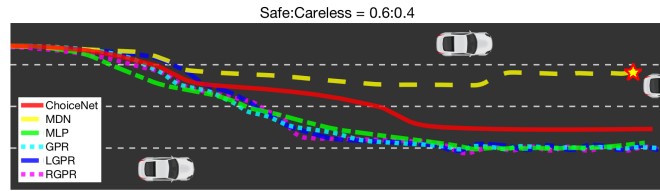

Figure 5: Resulting trajectories of compared methods trained with mixed demonstrations. (best viewed in color).

Table 5: Collision rates of compared methods on straight lanes.

| Outliers | ChoiceNet | MDN | MLP | GPR | LGPR | RGPR |
|---|---|---|---|---|---|---|
| 0% | **0**% | 50.83% | **0**% | 0.83% | 4.17% | 3.33% |
| 10% | **0**% | 38.33% | **0**% | 2.5% | 1.67% | 4.17% |
| 20% | **0**% | 41.67% | **0**% | 7.5% | 6.67% | 10% |
| 30% | **0**% | 66.67% | 1.67% | 4.17% | 1.67% | 7.5% |
| 40% | **0.83**% | 35% | 3.33% | 6.67% | 6.67% | 24.17% |

Table 6: Root mean square lane deviation distances (m) of compared methods on straight lanes.

| Outliers | ChoiceNet | MDN | MLP | GPR | LGPR | RGPR |
|---|---|---|---|---|---|---|
| 0% | 0.314 | 0.723 | **0.300** | 0.356 | 0.349 | 0.424 |
| 10% | **0.352** | 0.387 | 0.438 | 0.401 | 0.446 | 0.673 |
| 20% | **0.349** | 0.410 | 0.513 | 0.418 | 0.419 | 0.725 |
| 30% | **0.368** | **0.368** | 0.499 | 0.455 | 0.476 | 0.740 |
| 40% | **0.370** | 0.574 | 0.453 | 0.453 | 0.453 | 0.636 |

policy from driving demonstrations collected from both safe and careless driving modes. We use the same set of methods used for the previous task. The policy function is defined as a mapping between four dimensional input features consist of three frontal distances to left, center, and right lanes and lane deviation distance from the center of the lane to the desired heading. Once the desired heading is computed, the angular velocity of a car is computed by $10 * (\theta_{\text{desired}} - \theta_{\text{current}})$ and the directional velocity is fixed to $10m/s$. The driving demonstrations are collected from keyboard inputs by human users. The objective of this experiment is to assess its performance on a training set generated from two different distributions. We would like to note that this task does not have a reference target function in that all demonstrations are collected manually. Hence, we evaluated the performances of the compared methods by running the trained policies on a straight track by randomly deploying static cars.

Table 5 and Table 6 indicate collision rates and RMS lane deviation distances of the tested methods, respectively, where the statistics are computed from $50$ independent runs on the straight lane by randomly placing static cars as shown in Figure 5. ChoiceNet clearly outperforms compared methods in terms of both safety (low collision rates) and stability (low RMS lane deviation distances).

Here, we describe the features used for the autonomous driving experiments. As shown in the manuscript, we use a four dimensional feature, a lane deviation distance of an ego car, and three frontal distances to the closest car at left, center, and right lanes as shown in Figure 6. We upperbound the frontal distance to $40m$. Figure 7(a) and 7(b) illustrate manually collected trajectories of a safe driving mode and a careless driving mode.

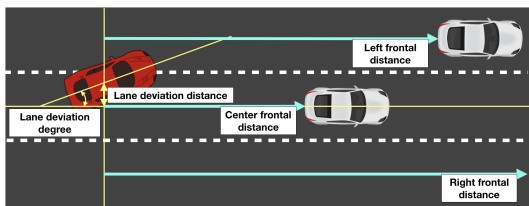

Figure 6: Descriptions of the featrues of an ego red car used in autonomous driving experiments.

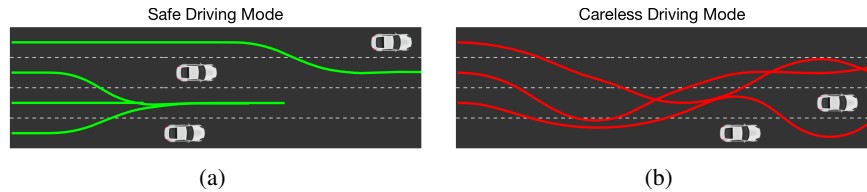

Figure 7: Manually collected trajectories of (a) safe driving mode and (b) careless driving mode. (best viewed in color).

## B.2 CLASSIFICATION TASKS

### B.2.1 ABLATION STUDY ON MNIST

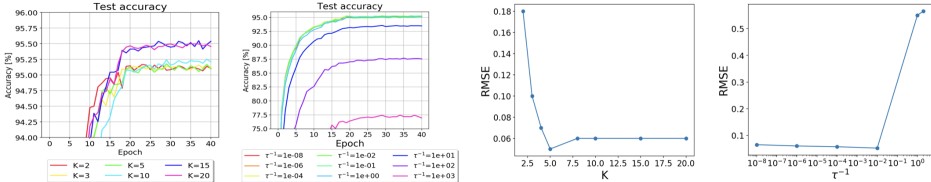

Above figures show the results of ablation study when varying the number of mixture $K$ and the expected measurement variance $\tau^{-1}$. Left two figures indicate test accuracies using the MNIST dataset where $90\%$ of train labels are randomly shuffled and right two figures are RMSEs using a synthetic one-dimensional regression problem in Section 4.1. We observe that having bigger $K$ is beneficial to the classification accuracies. In fact, the results achieved here with $K$ equals 15 and 20 are better than the ones reported in the submitted manuscript. $\tau^{-1}$ does not affect much unless it is exceedingly large.

### B.2.2 DIFFERENT TYPES OF NOISE ON MNIST

Here, we present additional experimental results using the MNIST dataset on following three different scenarios:

1. Biased label experiments where we randomly assign the percentage of the training labels to label $0$.

2. Random shuffle experiments where we randomly replace the percentage of the training labels from the uniform multinomial distribution.

3. Random permutation experiments where we replace the percentage of the labels based on the label permutation matrix where we follow the random permutation in (Reed et al., 2014).

The best and final accuracies on the intact test dataset for biased label experiments are shown in Table 7. In all corruption rates, ChoiceNet achieves the best performance compared to two baseline methods. The learning curves of the biased label experiments are depicted in Figure 8. Particularly, we observe unstable learning curves regarding the test accuracies of ConvNet and Mixup. As training accuracies of such methods show stable learning behaviors, this can be interpreted as the networks are simply memorizing noisy labels. In the contrary, the learning curves of ChoiceNet show stable behaviors which clearly indicates the robustness of the proposed method.

Table 7: Test accuracies on the MNIST dataset with biased label.

| Corruption $p$ | Configuration | Best | Last |
|---|---|---|---|
| 25% | ConvNet | 95.4 | 89.5 |
| | ConvNet+Mixup | 97.2 | 96.8 |
| | ChoiceNet | **99.2** | **99.2** |
| 40% | ConvNet | 86.3 | 76.9 |
| | ConvNet+Mixup | 87.2 | 87.2 |
| | ChoiceNet | **98.2** | **97.6** |
| 45% | ConvNet | 76.1 | 69.8 |
| | ConvNet+Mixup | 74.7 | 74.7 |
| | ChoiceNet | **94.7** | **89.0** |
| 47% | ConvNet | 72.5 | 64.4 |
| | ConvNet+Mixup | 69.2 | 68.2 |
| | ChoiceNet | **88.5** | **80.0** |

Table 8: Test accuracies on the MNIST dataset with corrupt label.

| Corruption $p$ | Configuration | Best | Last |
|---|---|---|---|
| 50% | ConvNet | 97.1 | 95.9 |
| | ConvNet+Mixup | 98.0 | 97.8 |
| | ChoiceNet | **99.1** | **99.0** |
| 80% | ConvNet | 90.6 | 79.0 |
| | ConvNet+Mixup | 95.3 | 95.1 |
| | ChoiceNet | **98.3** | **98.3** |
| 90% | ConvNet | 76.1 | 54.1 |
| | ConvNet+Mixup | 78.6 | 42.4 |
| | ChoiceNet | **95.9** | **95.2** |
| 95% | ConvNet | 50.2 | 31.3 |
| | ConvNet+Mixup | 53.2 | 26.6 |
| | ChoiceNet | **84.5** | **66.0** |

The experimental results and learning curves of the random shuffle experiments are shown in Table 8 and Figure 9. The convolutional neural networks trained with Mixup show robust learning behaviors when 80% of the training labels are uniformly shuffled. However, given an extremely noisy dataset (90% and 95%), the test accuracies of baseline methods decrease as the number of epochs increases. ChoiceNet shows outstanding robustness to the noisy dataset in that the test accuracies do not drop even after 50 epochs for the cases where the corruption rates are below 90%. For the 95% case, however, over-fitting is occured in all methods.

Table 9 and Figure 10 illustrate the results of the random permutation experiments. Specifically, we change the labels of randomly selected training data using a permutation rule: $(0, 1, 2, 3, 4, 5, 6, 7, 8, 9) \rightarrow (7, 9, 0, 4, 2, 1, 3, 5, 6, 8)$ following (Reed et al., 2014). We argue that this setting is more arduous than the random shuffle case in that we are intentionally changing the labels based on predefined permutation rules.

### B.2.3    CIFAR-10

Here, we present detailed learning curves of the CIFAR-10 experiments while varying the noise level from 20% to 80% following the configurations in (Zhang et al., 2017).

Table 9: Test accuracies on the MNIST dataset with randomly permuted label.

| Corruption $p$ | Configuration | Best | Last |
|---|---|---|---|
| 25% | ConvNet | 94.4 | 92.2 |
| | ConvNet+Mixup | 97.6 | 97.6 |
| | ChoiceNet | **99.2** | **99.2** |
| 40% | ConvNet | 77.9 | 71.8 |
| | ConvNet+Mixup | 84.0 | 83.0 |
| | ChoiceNet | **99.2** | **98.8** |
| 45% | ConvNet | 68.0 | 61.4 |
| | ConvNet+Mixup | 68.9 | 55.8 |
| | ChoiceNet | **98.0** | **97.1** |
| 47% | ConvNet | 58.2 | 53.9 |
| | ConvNet+Mixup | 60.2 | 53.4 |
| | ChoiceNet | **92.5** | **86.1** |

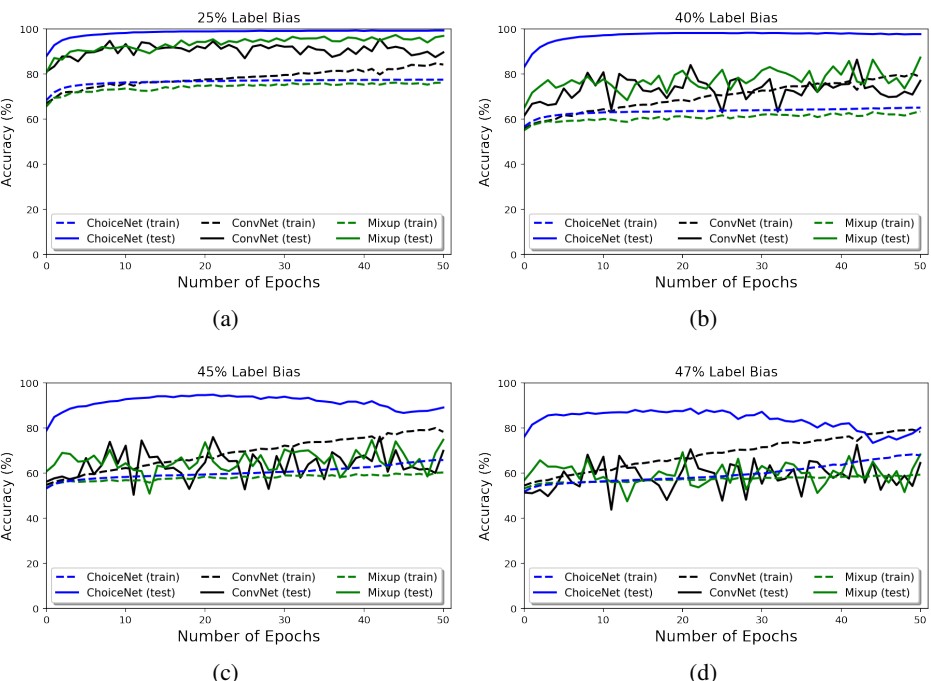

(a)  (b)

(c)  (d)

Figure 8: Learning curves of compared methods on random bias experiments using MNIST with different noise levels.

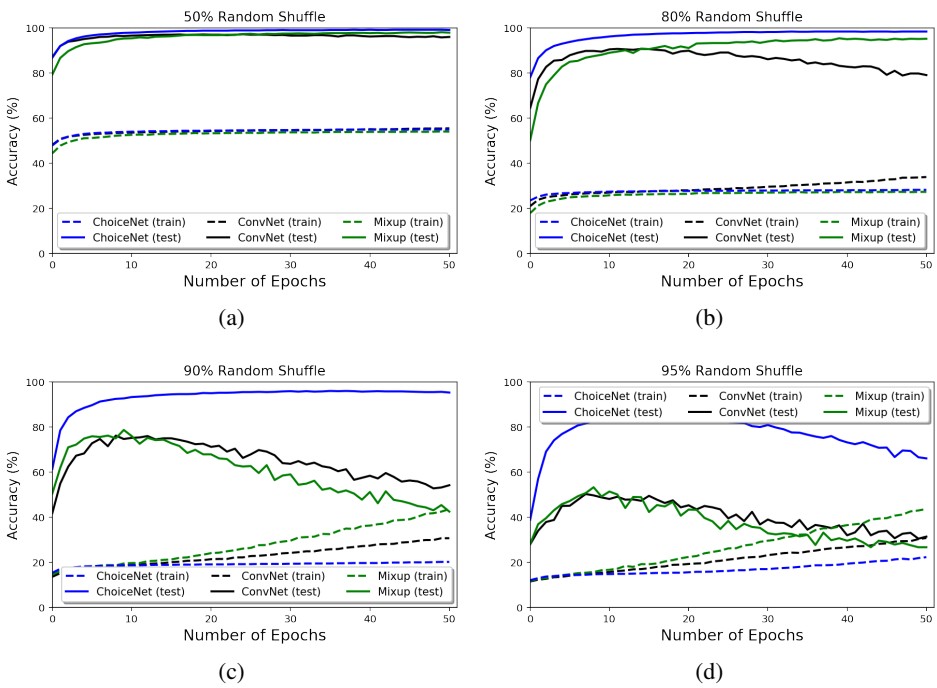

Figure 9: Learning curves of compared methods on random shuffle experiments using MNIST with different noise levels.

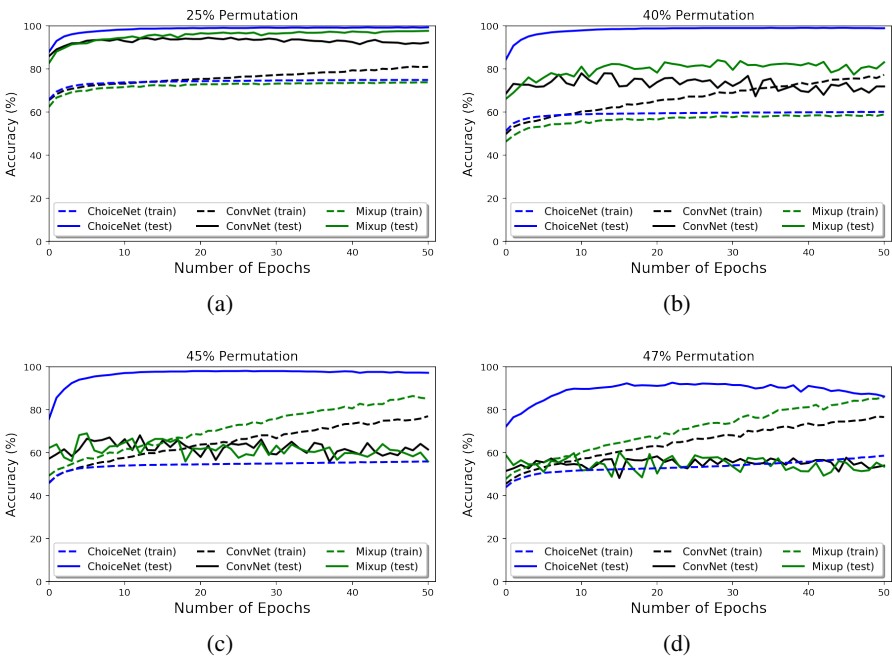

Figure 10: Learning curves of compared methods on random permutation experiments using MNIST with different noise levels.

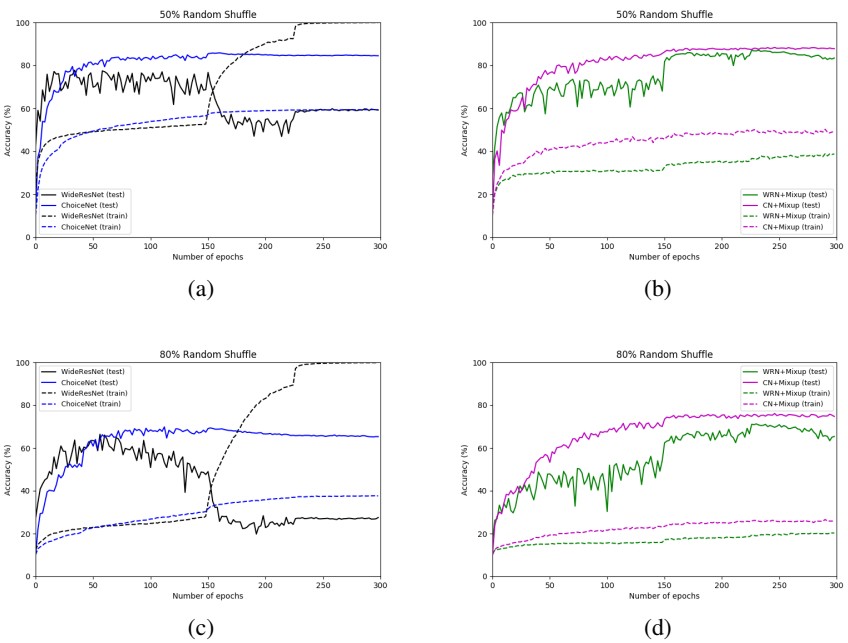

Figure 11: Learning curves of compared methods on CIFAR-10 experiments with different noise levels.

