# OpenReview forum: "ChoiceNet: Robust Learning by  Revealing Output Correlations"
_ICLR.cc/2019/Conference_

### Official Review · AnonReviewer3 · 2018-11-01
**ok papers but lacking of related works, important baselines and well-motivated storyline.**

**Rating:** 5
**Confidence:** 5

**Review:**

This paper formulates a new deep learning method called ChoiceNet for noisy data. Their main idea is to estimate the densities of data distributions using a set of correlated mean functions. They argue that ChoiceNet can robustly infer the target distribution on corrupted data.

Pros:

1. The authors find a new angle for learning with noisy labels. For example, the keypoint of ChoiceNet is to design the mixture of correlated density network block.

2. The authors perform numerical experiments to demonstrate the effectiveness of their framework in both regression tasks and classification tasks. And their experimental result support their previous claims.

Cons:

We have three questions in the following.

1. Related works: In deep learning with noisy labels, there are three main directions, including small-loss trick [1-3], estimating noise transition matrix [4-6], and explicit and implicit regularization [7-9]. I would appreciate if the authors can survey and compare more baselines in their paper instead of listing some basic ones.

2. Experiment:
2.1 Baselines: For noisy labels, the authors should add MentorNet [1] as a baseline https://github.com/google/mentornet From my own experience, this baseline is very strong. At the same time, they should compare with VAT [7].

2.2 Datasets: For datasets, I think the author should first compare their methods on symmetric and aysmmetric noisy data [4]. Besides, the current paper only verifies on vision datasets. The authors are encouraged to conduct 1 NLP dataset.

3. Motivation: The authors are encouraged to re-write their paper with more motivated storyline. The current version is okay but not very exciting for idea selling.

References:

[1] L. Jiang, Z. Zhou, T. Leung, L. Li, and L. Fei-Fei. Mentornet: Learning data-driven curriculum for very deep neural networks on corrupted labels. In ICML, 2018.

[2] M. Ren, W. Zeng, B. Yang, and R. Urtasun. Learning to reweight examples for robust deep learning. In ICML, 2018.

[3] B. Han, Q. Yao, X. Yu, G. Niu, M. Xu, W. Hu, I. Tsang, M. Sugiyama. Co-teaching: Robust training of deep neural networks with extremely noisy labels. In NIPS, 2018.

[4] G. Patrini, A. Rozza, A. Menon, R. Nock, and L. Qu. Making deep neural networks robust to label noise: A loss correction approach. In CVPR, 2017.

[5] J. Goldberger and E. Ben-Reuven. Training deep neural-networks using a noise adaptation layer. In ICLR, 2017.

[6] S. Sukhbaatar, J. Bruna, M. Paluri, L. Bourdev, and R. Fergus. Training convolutional networks with noisy labels. In ICLR workshop, 2015.

[7] T. Miyato, S. Maeda, M. Koyama, and S. Ishii. Virtual adversarial training: A regularization method for supervised and semi-supervised learning. ICLR, 2016.

[8] A. Tarvainen and H. Valpola. Mean teachers are better role models: Weight-averaged consistency targets improve semi-supervised deep learning results. In NIPS, 2017.

[9] S. Laine and T. Aila. Temporal ensembling for semi-supervised learning. In ICLR, 2017.

---

> ### Author Response · Authors · 2018-11-26
> **Modification on related work + more experiments (2/2)**
>
> 3. Motivation:
>
> As reviewer 1 and 3 pointed out, the manuscript requires more explanations regarding the proposed methods, Cholesky transform and MCDN block. Let us brieﬂy explain the motivations (backgrounds) and the practical meanings of the proposed methods. We will add them to the revised version. (We didn’t modify this part of the manuscript yet).
>
> 1. To handle noisy data, we reveal the quality of each data using the notion of correlation between output features. Specifically, we model the data to be collected from a mixture of a target distribution p(y|x) and other irrelevant distributions q(y|x). We quantify the irrelevancy (or independency) by correlation ρ between p(y|x) and q(y|x) where ρ ∈ [−1, 1]. Intuitively speaking, corrupted data will be modeled to be collected from a class of q(y|x) with small rho e.g. ρ = 0.
>
> 2. We model the target conditional distribution p(y|x) using a parametrized distribution with expected measurement variance τ −1 , i.e., p(y|x; θ) = N(y; f θ (x), τ −1 ) where f θ (·) is a neural network and θ is a set of parameters including µ_W and Σ_W . The Cholesky transform is proposed to construct a ρ-correlated conditional distribution using θ and ρ. In other words, the correlation between q(y|x; θ) (constructed from the Cholesky transform) and p(y|x; θ) is ρ.
>
> 3. Now, we can construct a mixture model of the target distribution, p(y|x; θ), and other distributions, q ρ (y|x; θ) parametrized by θ and the correlation parameter ρ. However, we still need to assess the quality (correlation) of each data point. Since the correlation information is not explicitly given, we model the correlation of each data to be a function of an input x, i.e., ρ φ (x), parametrized by φ and jointly optimize φ and θ using a mixture distribution. The mixture of correlated density network (MCDN) block is proposed for this purpose.
>
>
> [1] L. Jiang, Z. Zhou, T. Leung, L. Li, and L. Fei-Fei. Mentornet: Learning data-driven curriculum for very deep neural networks on corrupted labels. In ICML, 2018.
> [2] T. Miyato, S. Maeda, M. Koyama, and S. Ishii. Virtual adversarial training: A regularization method for supervised and semi-supervised learning. ICLR, 2016.
> [3] B. Han, Q. Yao, X. Yu, G. Niu, M. Xu, W. Hu, I. Tsang, M. Sugiyama, “Co-teaching: Robust Training of Deep Neural Networks with Extremely Noisy Labels”, NIPS, 2018.
> [4] G. Patrini, A. Rozza, A. Menon, R. Nock, and L. Qu. Making deep neural networks robust to label noise: A loss correction approach. In CVPR, 2017.
> [5] Y. Bengio, R. Ducharme, P. Vincent, C. Jauvin. A Neural Probabilistic Language Model. Journal of Machine Learning Research, 3:1137-1155, 2003.

---

> ### Author Response · Authors · 2018-11-26
> **Modification on related work + more experiments (1/2)**
>
> We appreciate the reviewer for the valuable reviews.
>
> 1. Related work: We admit that the current manuscript lacks comprehensive curation of related work. We rewrote the whole related work section and categorized existing work into four groups and try to compare them in a more principled way. Please refer to the revised manuscript.
>
> 2. Experiments: Following the review, we conducted three additional experiments: a) more baselines (MentorNet and VAT), b) using both symmetric and asymmetric noisy data, and c) using an NLP dataset.
>
> a). More baselines to current CIFAR-10 experiments: We implemented MentorNet [1] and VAT [2] to better evaluate the performance of the proposed method on current CIFAR-10 setting.
>
> corruption rate     20%     50%      80%
> ----------------------------------------------
> MentorNet PD        64.0%   49.0%    21.4%
> MentorNet DD        62.0%   43.1%    21.8%
> VAT                 82.0%   71.6%    16.9%
> ----------------------------------------------
> CN                  90.3%   84.6%    65.2%
> CN+Mixup            92.3%   87.9%    75.4%
>
> In all cases, the proposed methods (CN and CN+Mixup) outperforms the baselines.
>
>
> b) Asymmetric noise experiments following Co-teaching [3]. We implement the 9-layer CNN architecture following VAT [2] and Co-teaching [3] to fairly evaluate the performance of CIFAR10 experiments with both symmetric and asymmetric noise settings: Pair-45%, Symmetry-50%, and Symmetry-20%, using the authors’ implementations available on github. We also set other configurations such as having no data augmentation and activation functions to be the same as [3].
>
> (Single-run, last validation accuracy)
>               Pair-45%  sym-50%   sym-20%
> ------------------------------------------
> ChoiceNet     70.3%     85.2%     91.0%
> ------------------------------------------
> MentorNet     58.14%    71.10%    80.76%
> Co-teaching   72.62%    74.02%    82.32%
> F-correction  6.61%     59.83%    84.55%
>
> The results of MentorNet [1], Co-teaching [3], and F-correction [4] are copied from [3]. While our proposed method outperforms all compared methods on symmetric noise settings, it shows inferior performances to Co-teaching. This shows the weakness of the proposed method. In other words, our mixture distribution failed to correctly infer the dominant distribution which shows the weakness of the mixture-based method. However, we would like to note that Co-teaching [3] is complementary to our method where one can combine these two methods by using two ChoiceNets and update each network using Co-teaching.
>
> c) Natural language processing experiments: We used a Large Movie Review Dataset consist of 25,000 movie reviews for training and 25,000 reviews for testing. Each movie review (sentences) is mapped to a 128-dimensional feature vector using feed-forward Neural-Net Language Models [5] and we tested the robustness of the proposed method, mix-up, and naive MLP baseline by randomly flipping the labels.
>
> random flip rate    0%      10%     20%     30%     40%
> -------------------------------------------------------------
> ChoiceNet           79.43%  79.50%  78.66%  77.10%  73.98%
> Mix-up              79.77%  78.73%  77.58%  75.85%  69.63%
> Baseline (MLP)      79.04%  77.88%  75.70%  69.05%  62.83%
> VAT                 76.40%  72.50%  69.20%  65.20%  58.30%

---

### Official Review · AnonReviewer2 · 2018-11-03
**Interesting Approach with Nice Results**

**Rating:** 6
**Confidence:** 4

**Review:**

The paper presents a framework, called ChoiceNet, for learning when the
supervision outputs (e.g., labels) are corrupted by noise. The method relies on
estimating the correlation between the training data distribution and a
target distribution, where training data distribution is assumed to be a mixture
of that target distribution and other unknown distributions. The paper also
presents some compelling results on synthetic and real datasets, for both
regression and classification problems.

The proposed idea builds on top of previously published work on Mixture Density
Networks (MDNs) and Mixup (Zhang et al, 2017). The main difference is the MDN
are modified to construct the Mixture of Correlated Density Network (MCDN)
block, that forms the main component of ChoiceNets.

I like the overall direction and idea of modelling correlation between the
target distribution and the data distribution to deal with noisy labels. The
results are also compelling and I thus lean towards accepting this paper. My
decision on "marginal accept" is based primarily on my unfamiliarity with this
specific area and that some parts of the paper are not very easy or intuitive
to read through.

== Related Work ==

I like the related work discussion, but would emphasize more the connection to
MDNs and to Mixup. Only one sentence is mentioned about Mixup but reading
through the abstract and the introduction that is the first paper that came to
my mind and thus I believe that it may deserve a bit more discussion.

Also, there are a couple more papers that felt relevant to this work but are
not mentioned:
  - Estimating Accuracy from Unlabeled Data: A Bayesian Approach, Platanios et al., ICML 2016.
    I believe this is related in how noisy labels are modeled (i.e., section 3
    in the reviewed paper) and in the idea of correlation/consistency as a means
    to detect errors. There are couple more papers in this line of work that
    may be relevant.
  - ADIOS: Architectures Deep In Output Space, Al-Shedivat et al., ICML 2016.
    I believe this is related in learning some structure in the output space,
    even though not directly dealing with noisy labels.

== Method ==

I believe the methods section could have been written in a more
clear/easy-to-follow way, but this may also be due to my unfamiliarity with this
area. Figure 1 is hard to parse and does not really offer much more than section
3.2 currently does. If the figure is improved with some more text/labels on
boxes rather than plain equations, it may go a long way in making the methods
section easier to follow.

I would also point out MCDN as the key contribution of this paper as ChoiceNet
is just any base network with an MCDN block stacked on top of this. Thus, I
believe this should be emphasized more to make your key contribution clear.

== Experiments ==

The experiments are nicely presented and are quite thorough. A couple minor
comments I have are:

  - It would be nice to run regression experiments for bigger real-world
    datasets, as the ones used seem to be quite small.
  - I am a bit confused at the fact that in table 3 you compare your method to
    mixup and in table 4 you also show results when using both your method and
    mixup combined. Up until that point I thought that mixup was posed as an
    alternative method, but here it seems it's quite orthogonal and can be used
    together, which I think makes sense, but would be good to clarify. Also,
    given that you show combined results in table 4, why not also perform
    exactly the same analysis for table 3 and also show numbers for CN + Mixup?

It would also be nice to use the same naming scheme for both tables. I would
use: ConvNet, ConvNet + CN, ConvNet + CN + Mixup, and the same with WRN for
table 4. This would make the tables easier to read because currently the first
thing that comes to mind is what may be different between the two setups given
that they are presented side-by-side but use different naming conventions.

One question that comes to mind is that you make certain assumptions on the
kinds of noise your model can capture, so are there any cases where you have
good intuition as to why your model may fail? It would be good to present a
short discussion on this to help readers understand whether they can benefit by
using your model or not.

---

> ### Author Response · Authors · 2018-11-26
> **More experiments + modified related work + limitations (2/2)**
>
> We conducted additional experiments based on other reviews where we observe that the proposed method show superior performance to symmetric noises but vulnerable to asymmetric noise on CIFAR-10 following the settings in [3]. We implement the 9-layer CNN architecture following VAT [5] and Co-teaching [3] to fairly evaluate the performance of CIFAR10 experiments with both symmetric and asymmetric noise settings: Pair-45%, Symmetry-50%, and Symmetry-20%, using the authors’ implementations available on github. Pair-45% flips 45% of each label to the next label. For example, randomly flipping 45% of label 1 to label 2 and label 2 to label3. On the other hand, Symmetriy-50% randomly assigns 50% of each label to other labels uniformly. For example, Symmetriy-50% randomly flips 50% the labels of instances whose original label is 1 to a random label sampled from 2-10.
>
> We set other configurations such as the network topology and an activation functions to be the same as [3].
>
> (Single-run, last validation accuracy)
>               Pair-45%  sym-50%   sym-20%
> ------------------------------------------
> ChoiceNet     70.3%     85.2%     91.0%
> ------------------------------------------
> MentorNet     58.14%    71.10%    80.76%
> Co-teaching   72.62%    74.02%    82.32%
> F-correction  6.61%     59.83%    84.55%
>
> The results of MentorNet [6], Co-teaching [3], and F-correction [7] are copied from [3]. While our proposed method outperforms all compared methods on symmetric noise settings, it shows inferior performances to Co-teaching. This shows the weakness of the proposed method. In other words, our mixture distribution failed to correctly infer the dominant distribution which shows the weakness of the mixture-based method. However, we would like to note that Co-teaching [5] is complementary to our method where one can combine these two methods by using two ChoiceNets and update each network using Co-teaching.
>
> * We also conducted additional experiments to show the strength of the proposed method.
>
> a). More baselines to current CIFAR-10 experiments: We implemented MentorNet [6] and VAT [5] to better evaluate the performance of the proposed method on current CIFAR-10 setting.
>
> corruption rate     20%     50%      80%
> ----------------------------------------------
> MentorNet PD        64.0%   49.0%    21.4%
> MentorNet DD        62.0%   43.1%    21.8%
> VAT                 82.0%   71.6%    16.9%
> ----------------------------------------------
> CN+Mixup            92.3%   87.9%    75.4%
>
> b) Natural language processing experiments: We used a Large Movie Review Dataset consist of 25,000 movie reviews for training and 25,000 reviews for testing. Each movie review (sentences) is mapped to a 128-dimensional feature vector using feed-forward Neural-Net Language Models [8] and we tested the robustness of the proposed method, mix-up, and naive MLP baseline by randomly flipping the labels.
>
> random flip rate    0%      10%     20%     30%     40%
> -------------------------------------------------------------
> ChoiceNet           79.43%  79.50%  78.66%  77.10%  73.98%
> Mix-up              79.77%  78.73%  77.58%  75.85%  69.63%
> Baseline (MLP)      79.04%  77.88%  75.70%  69.05%  62.83%
> VAT                 76.40%  72.50%  69.20%  65.20%  58.30%
>
> Similar to regression experiments, ChoiceNet shows the superior performance in the presence of outliers where we observe that the proposed method can be used for NLP tasks as well.
>
> [1] V. Belagiannis, C. Rupprecht, G, Carneiro, N. Navab, "Robust Optimization for Deep Regression", ICCV, 2015
> [2] H. Zhang, M. Cisse, Y. Dauphin, D. Lopez-Paz, “mixup: Beyond Empirical Risk Minimization“, ICLR, 2018.
> [3] B. Han, Q. Yao, X. Yu, G. Niu, M. Xu, W. Hu, I. Tsang, M. Sugiyama, “Co-teaching: Robust Training of Deep Neural Networks with Extremely Noisy Labels”, NIPS, 2018.
> [4] Platanios, E. Antonios, A. Dubey, and T. Mitchell. "Estimating accuracy from unlabeled data: A bayesian approach." International Conference on Machine Learning. 2016.
> [5] T. Miyato, S. Maeda, M. Koyama, and S. Ishii. Virtual adversarial training: A regularization method for supervised and semi-supervised learning. ICLR, 2016.
> [6] L. Jiang, Z. Zhou, T. Leung, L. Li, and L. Fei-Fei. Mentornet: Learning data-driven curriculum for very deep neural networks on corrupted labels. In ICML, 2018.
> [7] G. Patrini, A. Rozza, A. Menon, R. Nock, and L. Qu. Making deep neural networks robust to label noise: A loss correction approach. In CVPR, 2017.
> [8] Y. Bengio, R. Ducharme, P. Vincent, C. Jauvin. A Neural Probabilistic Language Model. Journal of Machine Learning Research, 3:1137-1155, 2003.

---

> ### Author Response · Authors · 2018-11-26
> **More experiments + modified related work + limitations (1/2)**
>
> We thank the reviewer for the valuable comments, especially the suggestions regarding the related work. We admit that the current explanation about the proposed method is not straightforward and has some rooms for the improvements. Followings are the motivation of the proposed method and we will revise the manuscript so that the readers can better understand the concept more easily. (We didn’t modify this part of the manuscript yet).
>
> 1. To handle noisy data, we reveal the quality of each data using the notion of correlation between output features. Specifically, we model the data to be collected from a mixture of a target distribution p(y|x) and other irrelevant distributions q(y|x). We quantify the irrelevancy (or independency) by correlation ρ between p(y|x) and q(y|x) where ρ ∈ [−1, 1]. Intuitively speaking, corrupted data will be modeled to be collected from a class of q(y|x) with small rho e.g. ρ = 0.
>
> 2. We model the target conditional distribution p(y|x) using a parametrized distribution with expected measurement variance τ −1 , i.e., p(y|x; θ) = N(y; f θ (x), τ −1 ) where f θ (·) is a neural network and θ is a set of parameters including µ_W and Σ_W . The Cholesky transform is proposed to construct a ρ-correlated conditional distribution using θ and ρ. In other words, the correlation between q(y|x; θ) (constructed from the Cholesky transform) and p(y|x; θ) is ρ.
>
> 3. Now, we can construct a mixture model of the target distribution, p(y|x; θ), and other distributions, q ρ (y|x; θ) parametrized by θ and the correlation parameter ρ. However, we still need to assess the quality (correlation) of each data point. Since the correlation information is not explicitly given, we model the correlation of each data to be a function of an input x, i.e., ρ φ (x), parametrized by φ and jointly optimize φ and θ using a mixture distribution. The mixture of correlated density network (MCDN) block is proposed for this purpose.
>
> Following the reviewer’s comments, we conducted additional regression experiments using the Boston housing dataset. Here, we used the Boston housing price dataset and checked the robustness of the proposed method and compared with standard multi-layer perceptrons with four different types of loss functions: standard L2-loss, L1-loss which is known to be robust to outliers, a robust loss function proposed in [1], and a leaky robust function extending [1]. We further implement the leaky version of [1] in that the original loss function with Tukey’s biweight function discards the instances whose residuals exceed certain threshold.
>
> Outlier rate 	0%    5%    10%   15%   20%    30%    40%    50%
> -----------------------------------------------------------------------
> ChoiceNet       3.29  3.71  3.99  4.45  4.77   5.94   6.80   9.00
> L2 loss         3.22  4.61  5.97  6.65  7.51   9.04   9.88   10.92
> L1 loss         3.26  4.36  5.72  6.61  7.16   8.65   9.69   10.33
> Robust loss     4.28  4.63  6.36  6.59  8.08   10.54  10.94  11.96
> Leaky Robust    3.36  4.51  5.71  6.54  7.08   8.67   9.68   10.46
>
> We also modify the naming convention in the experiment section, e.g., ConvNet+CN+Mixup.In fact, we believe this naming convention can help understanding the benefit of the proposed method. In fact, it can be combined with other methods for achieving robustness such as mixup [2] or co-teaching [3] as these methods are compatible with our method.
> We rewrote the related work section to better categorize existing and current studies and added [4] to the related work.

---

### Official Review · AnonReviewer1 · 2018-11-08
**Interesting Approach with Insufficient Results**

**Rating:** 4
**Confidence:** 4

**Review:**

This paper presents an apparently original method targeted toward models training in the presence of low-quality or corrupted data. To accomplish this they introduce a "mixture of correlated density network" (MCDN), which processes representations from a backbone network, and the MCDN models the corrupted data generating process. Evaluation is on a regression problem with an analytic function, two MuJoCo problems, MNIST, and CIFAR-10.

This paper's primary strength is that the proposed method is a tool quite distinct from recent work, in that it does not use bootstrapping or solely use corruption transition matrices. The paper is typeset well. In addition to this, the experimentation has unusual breadth.

However, the synthetic regression task is a nice proof-of-concept, but thorough regression evaluation could perhaps include the Boston Housing Prices dataset or some UCI datasets.

The hamartia of this paper is that it does not provide sufficient depth in its computer vision experiments. For one, experimentation on CIFAR-100 would be appreciated.
In the CIFAR-10 experiments, they consider one label corruption setting and lack experimentation on uniform label corruptions.
The related works has thorough coverage on label corruption, but these works do not appear in the experiments. They instead compare their label corruption technique to mixup, a general-purpose network regularizer. It is not clear why it is thought the "state-of-the-art technique on noisy labels"; this may be true among network regularization approaches (such as dropout) but not among label correction techniques. For this problem I would expect comparison to at least three label correction techniques, but the comparison is to one technique which was not primarily designed for label corruption.


Nitpicks:
-In the related works we are told that a smaller learning rate can improve label corruption robustness. They train their method with a learning rate of 0.001; the baseline gets a learning rate of 0.1.
-The larger-than-usual batch size is 256 for their 22-4 Wide ResNets, and at the same time they do not use dropout (standard for WRNs of this width) and use less weight decay than is common. Is this because of mixup? If so why is the weight decay two orders of magnitude less for your approach compared to the baseline? How were these various atypical parameters chosen?
-They also use gradient clipping for their method, which is extremely rare for CIFAR-10 classification. Why is this necessary?
-This document could be cleaner by eschewing the Theorem of this paper, which "states that a correlation between two random matrices is invariant to an affine transform." For this audience, I suspect this theorem is unnecessary. Likewise the three lines expended for the maths of a Gaussian probability density function could probably be used for other parts of this paper.
-"a leverage optimization method which optimizes the leverage of each demonstrations is proposed. Unlike to former study," -> "a leverage optimization method which optimizes the leverage of each demonstration is proposed. Unlike a former study,"
-"In the followings," -> "In the following,"

Edit: The updated results need consistent baselines. For example, the method of [7] should be consistently compared against.

---

> ### Author Response · Authors · 2018-11-26
> **References**
>
> [1] B. Han, Q. Yao, X. Yu, G. Niu, M. Xu, W. Hu, Ivor W. Tsang, M. Sugiyama, “Co-teaching: Robust Training of Deep Neural Networks with Extremely Noisy Labels”, NIPS, 2018.
> [2] V. Belagiannis, C. Rupprecht, G. Carneiro, N. Navab, "Robust Optimization for Deep Regression", ICCV, 2015
> [3] Y. Bengio, R. Ducharme, P. Vincent, C. Jauvin. A Neural Probabilistic Language Model. Journal of Machine Learning Research, 3:1137-1155, 2003.
> [4] H. Zhang, M. Cisse, Y. Dauphin, D. Lopez-Paz, “mixup: Beyond Empirical Risk Minimization“, ICLR, 2018.
> [5] T. Miyato, S. Maeda, M. Koyama, and S. Ishii. Virtual adversarial training: A regularization method for supervised and semi-supervised learning. ICLR, 2016.
> [6] L. Jiang, Z. Zhou, T. Leung, L. Li, and L. Fei-Fei. Mentornet: Learning data-driven curriculum for very deep neural networks on corrupted labels. In ICML, 2018.
> [7] G. Patrini, A. Rozza, A. Menon, R. Nock, and L. Qu. Making deep neural networks robust to label noise: A loss correction approach. In CVPR, 2017.
> [8] P. Goyal, P. Dollár, R. Girshick, P. Noordhuis, “Accurate, Large Minibatch SGD: Training ImageNet in 1 Hour’, ArXiv, 2018.
> [9] K. He, et al. "Deep residual learning for image recognition.”, CVPR, 2016.

---

> ### Author Response · Authors · 2018-11-26
> **More experiments (robust regression, nlp, more baselines, and both symmetric and asymmetric noisy datasets) 2/2**
>
> 4. Asymmetric noise experiments following Co-teaching [1]. We implement the 9-layer CNN architecture following VAT [5] and Co-teaching [1] to fairly evaluate the performance of CIFAR10 experiments with both symmetric and asymmetric noise settings: Pair-45%, Symmetry-50%, and Symmetry-20%. Pair-45% flips 45% of each label to the next label. For example, randomly flipping 45% of label 1 to label 2. We used the authors’ implementations available on the github for generating the corrupted datasets. We also set configurations such as network topology (except for adding a MCDN layer instead of a linear layer), learning rate, optimizer max epoch to be the same as [1].
>
> (Single-run, last validation accuracy)
>               Pair-45%  sym-50%   sym-20%
> ------------------------------------------
> ChoiceNet     70.3%     85.2%     91.0%
> ------------------------------------------
> MentorNet     58.14%    71.10%    80.76%
> Co-teaching   72.62%    74.02%    82.32%
> F-correction  6.61%     59.83%    84.55%
>
> The results of MentorNet [6], Co-teaching [1], and F-correction [7] are copied from [1]. While our proposed method outperforms all compared methods on symmetric noise settings, it shows inferior performances to Co-teaching. This shows the weakness of the proposed method. In other words, our mixture distribution failed to correctly infer the dominant distribution which shows the weakness of the mixture-based method. However, we would like to note that Co-teaching [1] is complementary to our method where one can combine these two methods by using two ChoiceNets and update each network using Co-teaching.
>
> - We did not conduct CIFAR-100 experiments due to the limited time and computation resources available. However, we plan to do additional experiments following the settings from Co-teaching [1].
>
> Responses to nitpicks:
> -In the related works we are told that a smaller learning rate can improve label corruption robustness. They train their method with a learning rate of 0.001; the baseline gets a learning rate of 0.1.
> => The learning rate of 0.001 is only applied for the first epoch and the base learning rate of 0.1 is applied afterward. This technique is often called 'warming-up'. We will modify the manuscript so that there's no confusion about this [8,9].
>
> -The larger-than-usual batch size is 256 for their 22-4 Wide ResNets, and at the same time they do not use dropout (standard for WRNs of this width) and use less weight decay than is common. Is this because of mixup? If so why is the weight decay two orders of magnitude less for your approach compared to the baseline? How were these various atypical parameters chosen?
> => The optimal hyper-parameters of the proposed ChoiceNet varies from the standard resnet in that the way we train the network is totally different. The manual tuning of the hyper-parameters of both our method and baseline methods are automatically selected from the blackbox optimization method using a separate validation set.
>
> -They also use gradient clipping for their method, which is extremely rare for CIFAR-10 classification. Why is this necessary?
> => The gradient clipping is used to stabilize training. The main reason is that the proposed method first ‘samples’ a set of weights of the network and use the sampled parameters for inference. This seldom causes instability in the training phase.
>
> -This document could be cleaner by eschewing the Theorem of this paper, which "states that a correlation between two random matrices is invariant to an affine transform." For this audience, I suspect this theorem is unnecessary. Likewise the three lines expended for the maths of a Gaussian probability density function could probably be used for other parts of this paper.
> => We agree that current paper only uses a Gaussian prior distribution over the weight matrices. However, the theorem itself does not assume the Gaussian distribution. In fact, any centered distributions such as Gaussian or Laplacian can be used to model the weight matrices.
>
> Other typos will be modified in the revised manuscript.

---

> ### Author Response · Authors · 2018-11-26
> **More experiments (robust regression, nlp, more baselines, and both symmetric and asymmetric noisy datasets) 1/2**
>
> We thank the reviewer for the helpful comments. Especially, we agree that more in-depth experiments would be helpful for convincing the strength of the proposed method. In this regards, we conducted four additional experiments: a) robust regression experiments using a real-world dataset, b) experiments on NLP tasks, c) more baselines (MentorNet and VAT) on current CIFAR-10 experiments, and d) experiments with both symmetric and asymmetric following the recent work [1].
>
> a). Robust regression experiments: Here, we used the Boston housing price dataset and checked the robustness of the proposed method and compared our method with standard MLPs with four different types of loss functions: standard L2-loss, L1-loss which is known to be robust to outliers, a robust loss function proposed in [2], and a leaky robust function extending [2]. We implement the leaky version in that the Tukey’s biweight function discards the instances whose residuals exceed a certain threshold. Two-layer MLPs with 128 units and a relu activation is used for all scenarios. We vary the outlier ratio from 0% to 40% where the outputs of the outliers are uniformly sampled within the minimum and the maximum values of the training outputs. The results are as follows:
>
> outlier rate 	0%    5%    10%   15%   20%    30%    40%    50%
> -----------------------------------------------------------------------
> ChoiceNet       3.29  3.71  3.99  4.45  4.77   5.94   6.80   9.00
> L2 loss         3.22  4.61  5.97  6.65  7.51   9.04   9.88   10.92
> L1 loss         3.26  4.36  5.72  6.61  7.16   8.65   9.69   10.33
> Robust loss     4.28  4.63  6.36  6.59  8.08   10.54  10.94  11.96
> Leaky Robust    3.36  4.51  5.71  6.54  7.08   8.67   9.68   10.46
>
> The proposed method (ChoiceNet) outperforms all compared methods in the presence of outliers and shows a comparable performance without the outlier.
>
> 2. Natural language processing experiments: We used a Large Movie Review Dataset consist of 25,000 movie reviews for training and 25,000 reviews for testing. Each movie review (sentences) is mapped to a 128-dimensional embedding vector using feed-forward Neural-Net Language Models [3] and we tested the robustness of the proposed method, mix-up [4], VAT [5], and naive MLP baseline by randomly flipping the labels. In all experiments, we used two-layer MLPs with 128 hidden units and ReLU activations.
>
> random flip rate    0%      10%     20%     30%     40%
> -------------------------------------------------------------
> ChoiceNet           79.43%  79.50%  78.66%  77.10%  73.98%
> Mix-up              79.77%  78.73%  77.58%  75.85%  69.63%
> Baseline (MLP)      79.04%  77.88%  75.70%  69.05%  62.83%
> VAT                 76.40%  72.50%  69.20%  65.20%  58.30%
>
> Similar to regression experiments, ChoiceNet shows the superior performance in the presence of outliers where we observe that the proposed method can be used for NLP tasks as well.
>
> 3. More baselines to current CIFAR-10 experiments: We compared MentorNet [6] and VAT [5] to better evaluate the performance of the proposed method on the current CIFAR-10 setting. For MentorNet, we compare two methods: MentorNet PD which uses the pre-deﬁned curriculum to train StudentNet and the other, MentorNet DD, which uses data-driven curriculum to train StudentNet where we use Resent-101 for the StudentNet following the author’s implementations. We would like to note that the base networks of the StudentNet are not the same as the one we used for ChoiceNet, but even bigger. Due to the limited time for tuning the hyper-parameters, we simply used the existing implementations while changing the train and test dataset. To measure the robustness of compared methods, we vary the corruption probabilities from 20% to 80% and the results are as follows where the results of CN and CN+Mixup are copied from the current manuscript. We're working on reproducing MentorNet and VAT on the exact same network architecture.
>
> corruption rate     20%     50%      80%
> ----------------------------------------------
> MentorNet PD        64.0%   49.0%    21.4%
> MentorNet DD        62.0%   43.1%    21.8%
> VAT                 82.0%   71.6%    16.9%
> ----------------------------------------------
> CN                  90.3%   84.6%    65.2%
> CN+Mixup            92.3%   87.9%    75.4%
>
> In all cases, the proposed methods (CN and CN+Mixup) outperforms the baselines.

---

### Meta-Review · Area_Chair1 · 2018-12-15
**good direction but experiments lacking in some respects**

**Confidence:** 3
**Recommendation:** Reject

**Metareview:**

The paper addresses an interesting problem (learning in the presence of noisy labels) and provides extensive experiments. However, while the experiments in some sense cover a good deal of ground, reviewers raised issues with their quality, especially concerning baselines and depth (in terms of realism of the data). The authors provided many additional experiments during the rebuttal, but the reviewers did not find them sufficiently convincing.